# Modelling the effect of catena position and hydrology on soil chemical weathering

Vanesa García-Gamero[1], Tom Vanwalleghem[1], Adolfo Peña[2], Andrea Román-Sánchez[3], Peter A. Finke[4]

[1]Department of Agronomy, University of Córdoba, Da Vinci building, Madrid km 396 Rd., 14071 Córdoba, Spain
[2]Department of Rural Engineering, Civil Constructions and Engineering Projects, University of Córdoba, Da Vinci building, Madrid km 396 Rd., 14071 Córdoba, Spain.
[3]Department of Forest Ecology, The Silva Tarouca Research Institute, Lidická 25/27,602 00 Brno, Czech Republic.
[4]Department of Environment, University of Ghent, Coupure Links 653, 9000 Ghent, Belgium.

*Correspondence to*: Vanesa García-Gamero (g02gagav@uco.es)

**Abstract.** The sensitivity of chemical weathering to climatic and erosional forcing is well established at regional scales. However, soil formation is known to vary strongly along catenas where topography, hydrology, and vegetation cause differences in soil properties and possibly chemical weathering. This study applies the SoilGen model to evaluate the link between topographic position and hydrology with the chemical weathering of soil profiles on a north-south catena in southern Spain.

We simulated soil formation in seven selected locations over a 20000-year period and compared against field measurements. There was good agreement between simulated and measured chemical depletion fraction (CDF) ($R^2$=0.47). An important variation in CDF values along the catena was observed that is better explained by the hydrological variables than by the position along the catena alone or by the slope gradient. A positive trend between CDF data and soil moisture and infiltration and a negative trend with water residence time was found. This implies that these hydrological variables are good predictors of the variability of soil properties.

The model sensitivity was evaluated with a large precipitation gradient (200-1200 mm yr$^{-1}$). The model results show an increase of chemical weathering of the profiles up to a mean annual precipitation value of 800 mm yr$^{-1}$, after which it drops again. A marked depth gradient was obtained for CDF up to 800 mm yr$^{-1}$, and a uniform depth distribution was obtained with precipitation above this threshold. This threshold reflects a change in behaviour where the higher soil moisture and infiltration lead to shorter water transit times and decreased weathering. Interestingly, this corroborates similar findings on the relation of other soil properties to precipitation and should be explored in further research.

## 1 Introduction

The spatial variability of soil properties is conditioned by the five main soil forming factors: climate, organisms, relief, parent material, and time (Jenny, 1941). Differences in the spatial and temporal distribution of these factors cause both long and short-scale spatial heterogeneity. In recent years, different soil formation models have been developed that explain the landscape or large-scale soil variability well. Such models range from simple mechanistic soil depth models (Minasny & McBratney, 2001)

to more complex models that link different soil forming processes and erosion-deposition, for example, MILESD or LORICA (Temme and Vanwalleghem, 2016; Vanwalleghem et al., 2013).

However, the short-scale variability, or catena effect, has received much less attention. The interlocking of specific soil and vegetation associations at different landscape positions was first described by Milne (1935) and is widely used in soil science (Borden et al., 2020). The soil catena can be understood from the retention and movement of water and chemical elements linked to topography and vegetation (Reuter and Bell, 2001). Recently, Ferrier & Perron (2020) constructed a numerical model for the coevolution of topography, soils and soil mineralogy that allowed them to conclude that the hillslope scale has a critical

importance in the response of chemical weathering rates to changes in tectonics and climate. However, most existing soil formation models do not account for hydrology, nor for chemical weathering reactions. For example, the models marm3D (Cohen et al., 2010) and SSSPAM (Welivitiya et al., 2019) linked landscape and pedogenesis processes for catena spatial scales, but while they represent physical weathering and armouring well, they did not account for chemical weathering. At present, the only models of soil genesis with the capability of simulating water flow, physical and chemical weathering and

chemical equilibriums are one-dimensional, for example, SoilGen (Opolot et al., 2015). In spite of this limitation, such models have been applied successfully at the landscape scale, by modelling the different landscape positions independently. Finke (2012) for example modelled 3 soil profiles on different topographic positions in the loess belt of Belgium. Finke et al. (2013) modelled the spatial variation of soil horizons at 108 locations in a 1329 ha large forest area in the same loess belt region. However, there is still need to further test these one-dimensional models against field data from different environments, and

especially, to test their capabilities to model chemical weathering.

Field studies have shown the importance of chemical weathering in the overall soil formation processes, and have shown that physical erosion and chemical weathering are tightly coupled (Riebe et al., 2004) as the main processes in eroding environments. The combination of these processes determines the total denudation rate (Riebe et al., 2001). The contribution of chemical weathering (W) to the total denudation rate (D), W/D, can be inferred by comparing the concentration of immobile

elements in soil and bedrock, through the chemical depletion fraction (CDF) (Riebe et al., 2001).

There is an ongoing debate in the scientific community to whether chemical weathering is limited by physical erosion and the supply of fresh particles (e.g Larsen et al., 2014) or whether it is limited by reaction kinetics (e.g. Gabet & Mudd, 2009). Generally, it is assumed from models and field studies in different environments that soil weathering is supply-limited when erosion rates are low, and kinetically limited when erosion rates increase, as the shorter soil residence times imply that minerals

do not stay in the soil long enough to become fully weathered. Larsen et al. (2014) compiled data from the New Zealand Alps, among the world's most rapidly eroding mountain areas, and still found a positive relation between physical erosion rates and chemical weathering, indicating supply-limited conditions. On the other hand, is well established that climate and specifically water availability is also an important factor affecting chemical weathering rates (e.g. Maher, 2011). Climatic factors affect these rates significantly because of the dependency of the chemical weathering types of water to drive the chemical alteration

of rocks and are potentially accelerated by high temperatures because it affects the kinetics reactions and solubilities (Duarte et al., 2018). Schoonejans et al. (2016) confirmed the significant relation of chemical weathering to rainfall, by measuring

CDF along a climatic gradient in a semiarid environment in the Southern Betic Cordillera (Spain). Whereas much of soil weathering research has focused much on the critical control of physical erosion rates on chemical weathering, more recent research by Calabrese & Porporato (2020) stresses the importance of wetness. They suggest that water-limited environments are kinetic limited. Globally, they calculate 61% of the land to be kinetic-limited, while only 1% would be supply-limited. If their findings can be confirmed, it would imply that climatic conditions and soil hydrology are much more important than previously assumed. In any case, these authors point out that the factors affecting chemical weathering need to further be disentangled (Calabrese & Porporato, 2020).

However, the relation between weathering and water availability is not limited to precipitation and evapotranspiration but is also related to other factors such as infiltration or topography, and even vegetation. Along a catena, the hydrology is considerably different depending on the position (Chadwick et al., 2013). Dahlgren et al. (1997) in their work developed along an elevational transect in the Sierra Nevada in California (USA) found the maximum degree of chemical weathering with intermediate levels of precipitation and temperature. Riebe et al. (2004) made measurements on 42 study sites with highly variable climate regimes and they showed that the degree of chemical depletion increases systematically with temperature and precipitation. Mudd & Furbish (2006) presented a hillslope model that couples the evolution of topography and soil thickness by using immobile minerals, assessing the importance of hydrology on the rate of chemical weathering in hillslope soils. Yoo et al. (2007) developed a model that integrated the Riebe et al. (2001) method and tested it with experimental data from a watershed in south-eastern Australia for simultaneously quantifying the rates of chemical weathering and soil transport as a function of hillslope position. Afterward, this work was expanded by Yoo et al. (2009) who quantified soil chemical weathering rates along a grass-covered hillslope in the Tennessee Valley in Coastal California (USA) and began to elucidate the mechanisms that control the topographical dependence on chemical weathering being the next step includes hydrology into the model. Samouëlian & Cornu (2008) pointed out the role of water in soil formation and how the soil moisture regime variation and its influence on soil formation processes were not included in several models. Indeed, the soil moisture dynamics related to the topographical position has been studied by several authors, such as Salve et al. (2012), who have monitored this and other variables for 4 years with multiple measurement devices on a hillslope in coastal Mendocino County, in northern California, USA. Langston et al. (2011) explored the role of hydrology in saprolite formation through 2-D numerical calculations on two idealized slopes, north (NFS) and south-facing (SFS), respectively, in the Boulder Creek watershed, Colorado (USA). Highlighting the importance of time and hillslope aspect in the formation of the saprolite and the need to couple hydrologic models with reactive-transport models to better understand the distribution of chemical weathering intensity. Dixon et al. (2016), along 60 km of a north-south toposequence in the Waitaki Valley in the South Island of New Zealand, marked with an important precipitation gradient, identified a pedogenic threshold at mean annual precipitation of ~800 mm yr$^{-1}$. Braun et al.(2016) presented a model for the formation of regolith on geological timescales by chemical weathering, pointing out that this process has long been neglected in favor of hillslopes physical processes by geomorphologists, this process also being the mechanism that leads to the formation of aquifers from the unweathered bedrock (Lachassagne & Wyns, 2011). Brantley et al.(2017) developed a conceptual model to link chemical weathering reaction fronts to hillslope hydrology.

Knowing how difficult it is for hydrologists to select models, they developed a conceptual model to relate reaction fronts with water flow inside hillslopes, exemplified with field data for shale, granite, and diabase. The linear hill with two layers of lateral water flow is assumed to be characterized by reaction fronts. The upper and lower reaction fronts are separated by meters beneath hills on felsic rocks because of weathering-induced fracturing that permits water infiltration increases the regolith and reaction front thickness, and front spacing. In particular, groundwater removes highly soluble minerals, whereas interflow removes moderately soluble minerals. Previously, the weathering zone had been considered the one located over the groundwater level. However, Lebedeva & Brantley (2020) formulated a simplified weathering model to explore relationships between weathering and the water table taking into account the unsaturated and the saturated zone.

In response to the need to couple hydrology with chemical weathering to understand spatial variability in soil formation, this study applied the SoilGen model (Finke & Hutson, 2008) to different pedons and different hydrological conditions along a catena in a Mediterranean catchment in southern Spain. Finke & Hutson (2008) created the SoilGen model to reconstruct soil development based on present knowledge. It is a 1-D solute transport extended with various soil development processes such as physical and chemical weathering, clay migration, cycles of various elements including that of C and bioturbation, where soil forming-factors (climate, organisms, relief, parent material, and time) serve as initial- and boundary conditions to recreate soil formation over various parent materials.

The specific objectives of this study were: (i) to evaluate differences in simulated soil development between two opposing, north-south facing, hillslopes, from 20000 years before present to present; (ii) to relate soil development to differences in hydrological dynamics and (iii) to evaluate, through sensitivity analysis, the capability of the SoilGen model to simulate soil development in a Mediterranean catchment in granitic parent material.

## 2 Materials and methods

### 2.1 SoilGen model

SoilGen (Finke, 2012; Finke and Hutson, 2008) simulates the change of soil properties as a function of properties of the parent material and time-dependent drivers at the soil boundary (climate, vegetation, bioturbation, relief and deposition or erosion). The model operates at a typical spatial scale of 1 m$^2$, covers millenniums but takes dynamic time steps that vary per process (Fig. 1), depending on process speed. The flow of water, heat, gas and solutes is represented by numerical solutions to partial differential equations (Richards' equation, heat flow equation, gas diffusion equation, solute advection/dispersion equation), where soil profiles have 5-cm compartments. For water flow, the relationship between pressure head, water content and hydraulic conductivity are dynamically parameterized using a pedotransfer function based on the texture, organic matter content and bulk density. These properties are dynamically simulated per compartment: (i) The fate of organic carbon is simulated according to the concepts of the RothC26.3 model (Coleman & Jenkinson, 1996); (ii) The texture changes due to physical weathering of minerals; (iii) Clay migration and bioturbation affect the vertical distribution of all soil components; (iv) The bulk density varies because of mass gains/losses over the compartment.

Chemical weathering of minerals as well as organic matter decomposition release ions in the soil solution. These ions are distributed over precipitated, solution and exchange phases using a Gapon exchange mechanism and chemical equilibriums.

The model can simulate several agricultural practices and can also accommodate the removal (or addition) of top layers by erosion (or sedimentation, such as dust addition).

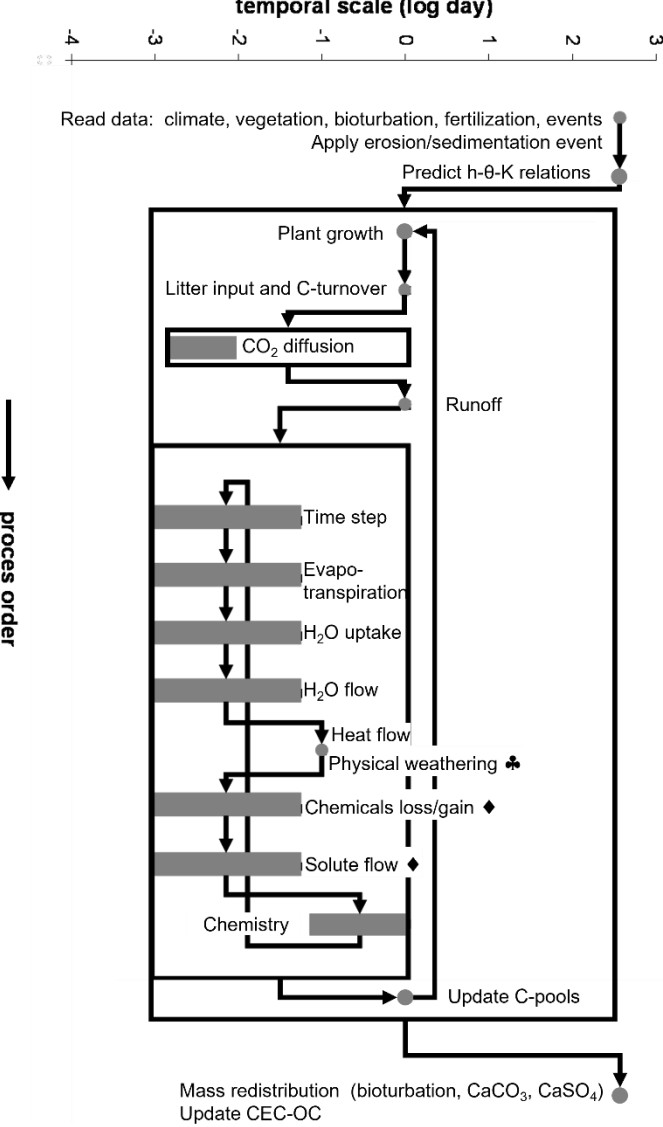

**Figure 1: Process flowchart of SoilGen. This figure was published in Quaternary International, 265, Peter A. Finke, Modeling the genesis of luvisols as a function of topographic position in loess parent material,3-17, Copyright Elsevier (2012).**

**2.2 Study site**

The study site is located in a semi-natural area of oak-woodland savanna or 'dehesa'(a traditional Mediterranean silvopastoral system, Olea & San Miguel-Ayanz, 2006) in Cardeña, within the Sierra Morena mountain range, in southern Spain (38.20° N; 4. 17º W, 700 m a.m.s.l.) (Fig. 2). The direction of the prevailing winds is southwest. Seven locations were chosen along a north-south catena, each separated at a distance between 50 and 100 m on two opposite hillslopes. A summary of the

characteristics of the locations is given in Table 1. A full soil profile description was made at each point, and soil samples were taken for chemical analysis up till the bedrock (see soil profile description Tables S1-S7 and physical-chemical characteristics Table S8 in the Supplement). Next, five soil moisture sensors were installed in each profile at different depths between 5 and 45 cm. For details on the soil hydrology dynamics, see García-Gamero et al. (2021) on two opposing hillslopes of a semi-arid, Mediterranean catchment in southern Spain, studied the interaction between hydrology, terrain and vegetation through soil

moisture, vegetation, and water table dynamics measurement, to quantify the aspect influence on ecohydrological dynamics of an oak-woodland savanna or 'dehesa'.

According to the Köppen-Geiger climate classification (Peel et al., 2007), the area has a continental Mediterranean climate (BSk) with an average annual rainfall of 878 mm (1981-2010), with cold winters and long, dry summers. The mean annual air temperature is 15.3 ºC, the coldest month is January, with a mean monthly temperature of 7ºC, and the hottest July, with a

mean monthly temperature of 25.4ºC (Carpintero et al., 2020).

Soils in the catchment are derived from Los Pedroches batholith parent material, which consists of a main granodioritic unit, several granite plutons, and an important acid-to-basic dike complex (Carracedo et al., 2009). These fall into three classifications: Regosols, Leptosols, and Cambisols according to the FAO-Unesco World Reference Base (IUSS Working Group WRB, 2015). The texture class ranges from sand to sandy loam with a soil depth generally ranging between 0.5 m along

the south aspect part of the transect and 1.0 m along the northern part of the transect ( Román-Sánchez et al., 2018), according to soil observations.

Vegetation in a dehesa includes sparse trees, holm (*Quercus Ilex* L.) and cork oaks (*Quercus suber*), shrubs, retama, and annual grasses such as *Lolium* sp., *Bromus* sp., and *Trifolium* sp., with maximum production in spring and a non-vegetative period in summer (Olea and San Miguel-Ayanz, 2006).

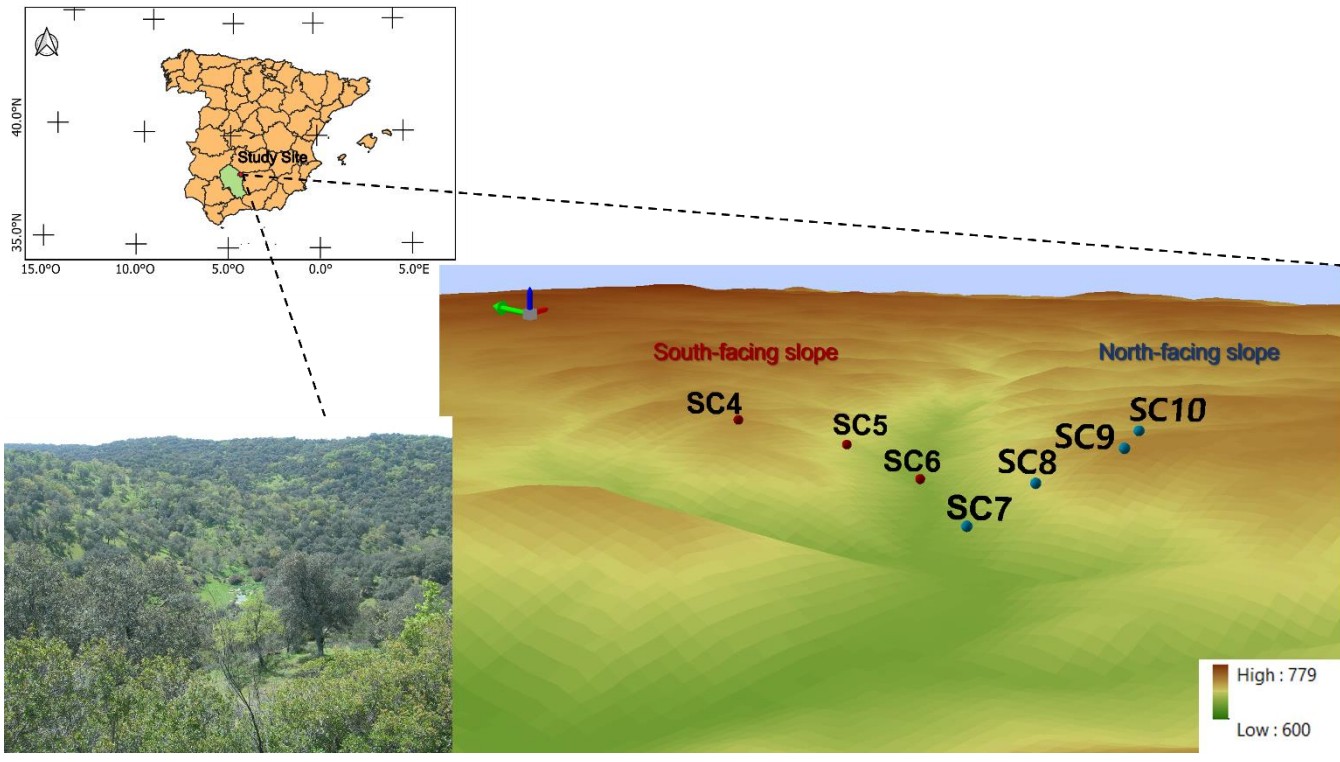

**Figure 2: General location map of the study area and a detailed terrain map showing the location of the seven analyzed soil profiles along the catena with two opposing north-south facing slopes (respectively NFS and SFS, blue and red dots). The inset photo shows a landscape view of the study area. Note the green arrow indicates the north.**

**Table 1. Characteristics of the seven locations (SC4-SC10) in the study site.**

| | SC4 | SC5 | SC6 | SC7 | SC8 | SC9 | SC10 |
|---|---|---|---|---|---|---|---|
| **Landform types[a]** | Summit | Backslope | Toeslope | Toeslope | Backslope | Shoulder | Summit |
| **Slope (°)** | 4 | 20 | 14 | 4 | 27 | 19 | 2 |
| **Upslope bearing (°)** | 35 | 27 | 40 | 55 | 61 | 66 | 53 |
| **Profile curvature (m$^{-1}$)** | -0.47 | -0.22 | 2.5 | 1.0 | -3.0 | 0.23 | -0.75 |
| **Plan curvature (m$^{-1}$)** | 0.78 | 0.73 | -0.39 | -0.9 | 4.24 | 0.67 | -0.65 |

[a]Following Schaetzl & Anderson (2005).

### 2.3 Testing of modelled soil hydrology

SoilGen simulates over millennium scales the flow of water, heat, solutes, and $CO_2$ in unconsolidated geomaterials by
numerically solving partial differential equations (the Richards equation, heat flow equation, advection-dispersion equation, and $CO_2$-diffusion equation respectively) (Finke & Hutson, 2008).

Given the importance of water fluxes for soil formation in general and chemical weathering specifically, the model was tested for its ability to simulate the soil moisture dynamics accurately. For this, the current soil profile characteristics in 7 locations were used to run the model over a two-year period in which soil moisture was measured, i.e., 2016 and 2017. The soil moisture
sensors, transmission line oscillator probes (model CS655, Campbell Scientific, Inc., Logan, UT), started to collect data at the end of 2016 so for the first part of the year the simulations were adapted using precipitation and evapotranspiration data from 2017. This could be considered a spin-up (because the initial soil moisture profile was not known).

To account for interception by vegetation, we reduced the daily precipitation by a fixed value of 2 mm, as suggested by Laio et al. (2001). These changes resulted in a reduction of the measured precipitation during the calibration period of 718.68 mm
to 525.40 mm.

Model-to-data correspondence was tested by comparing simulated values with measured values of soil moisture content, by the $R^2$ and the Nash-Sutcliffe Efficiency (NSE).

### 2.4 Model Inputs

The tested model was then run at the seven soil profile location for long-term simulations (20000 years). This simulation period represents the residence time of the soil profile based on field measurements with Optically-Stimulated Luminescence (Román-Sánchez et al., 2019). Table 2 summarizes the main model parameters and initial values. Figure 3 depicts the temporal variation of boundary inputs if time series were reconstructed over the simulation period. In order to represent correctly the Mediterranean climate variability, the climate data reconstructions were obtained by combining a mean temperature,
precipitation, or evapotranspiration from the pollen-based dataset by Davis et al.(2003), with the observed year-to-year variability of modern weather observations.


**Table 2. Inputs for the SoilGen Model.**

| Group | Input Variable or Parameter | Dimensions | As Initial Condition | Time series, in Typical Year | Time Series, Annual | Source for Data and/or Method |
|---|---|---|---|---|---|---|
| **Climate** | Temperature | °C | Yes | Weekly average and daily amplitude | January and July averages | Davis et al., 2003; Cardeña weather data |
| | Precipitation | mm | - | Daily depth, intensity, and chemical composition of rain | Annual sum | Davis et al., 2003; Cardeña weather data |
| | Potential evapotranspiration | mm | - | Weekly total | Annual sum | Hargreaves & Samani, 1985 |
| **Organisms** | Vegetation type | - | - | - | Vegetation type, rooting depth, and C input as litter | Olea & San Miguel-Ayanz, 2006; Unpublished data from Santa Clotilde 'dehesa' |
| | Bioturbation | $Mg\ ha^{-1}\ yr^{-1}$ | - | - | Yearly depth distribution | Gobat et al., 1998, p. 122 |
| **Relief** | Slope angle | degree | Yes | | | DEM; (García-Gamero et al., 2021) |
| | Slope aspect | degree | Yes | | | DEM; (García-Gamero et al., 2021) |
| | Wind direction | degree | Yes | | | Cardeña weather data |
| **Parent material** | Clay/Silt/Sand | Mass % | Yes | | | Granite: 7.0/17.4/75.6 Unpublished data from analyzed horizons in Santa Clotilde 'dehesa' |
| | OC | Mass % | Yes | | | 0.17 % |
| | Ca, Mg, Na, K, Al, SO$_4$, Cl, Alkalinity in solution | $mmol\ dm^{-3}$ | Yes | | | Unpublished data |
| | Ca, Mg, Na, K, Al, H on exchange complex and CEC | $mmol+\ kg^{-1}$ | Yes | | | Unpublished data from analyzed horizons in Santa Clotilde 'dehesa' |
| | CaCO$_3$/CaSO$_4$ | Mass % | Yes | | | 0.01/0% Unpublished data from analyzed horizons in Santa Clotilde 'dehesa' |
| | Gapon exchange coefficients | $(mol\ dm^{-3})^{1/n-1/m}$ | Yes | | | De Vries and Posch (2003) |
| | Ca, Mg, Na, K, Al in primary minerals | $mol^+\ kg^{-1}$ | Yes | | | Unpublished data from Santa Clotilde 'dehesa' |


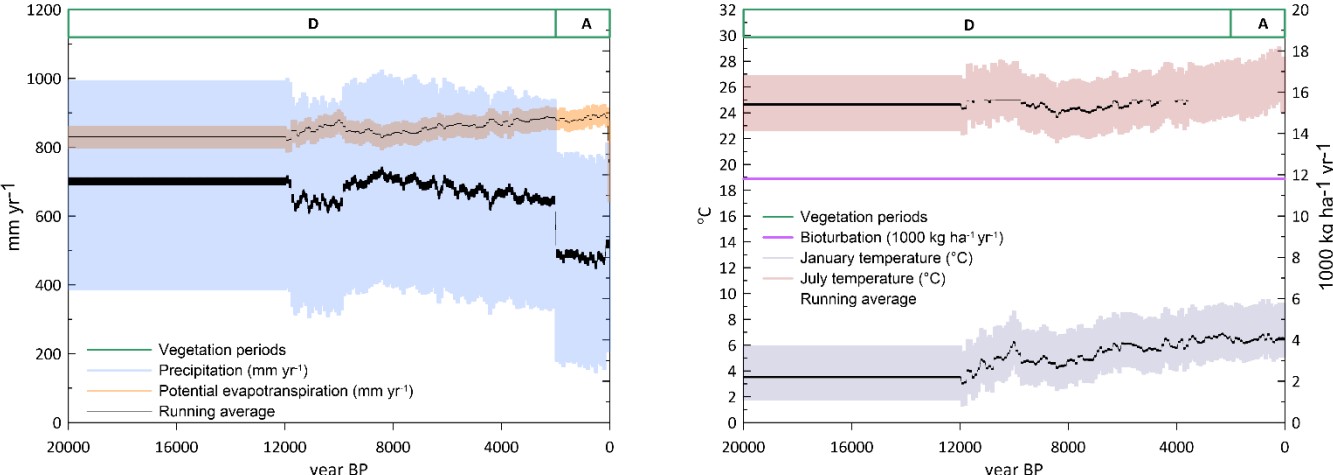

**Figure 3: Boundary conditions for the soil modelling, representing reconstructed climate and vegetation change over the last 20000 years, the left panel shows precipitation and evapotranspiration, the right panel shows temperature. The bandwidth indicates the year-to-year variability in the climate data, and the bold, black line indicates the 25 year running average. A= Agriculture, D= Deciduous wood.**

### 2.5 Chemical composition and weathering

Chemical depletion fraction (CDF), expressing the total fractional mass lost to chemical weathering, is a widely used indicator of the degree of chemical weathering of a soil profile (Dixon et al., 2009; Riebe et al., 2001):

$$CDF = 1 - \frac{Zr_r^{ROCK}}{Zr_r^{SOIL}}$$

where $Zr^{ROCK}$ is the Zr concentration in the bedrock and $Zr^{SOIL}$ is the Zr concentration in the soil. CDF values close to zero indicate the absence of weathering, as the Zr concentration of the soil is equal to that of the parent material. Values closer to 1 indicate higher chemical weathering, as the weathering of mobile elements results in a relative increase of the immobile Zr in the soil profile (Yoo et al., 2011).

CDF values were measured in the field by sampling each soil profile and comparing it to the minimum value registered in the soil profile (Table 3). The chemical analyzes were performed on a PHILIPS model PW 2404 wavelength-based X-ray fluorescence spectrometer (WDS) with a 4kW rhodium anode tube. The determination of the major elements was carried out from a fusion bead, with Lithium tetraborate and a flux/sample ratio of 10:1. Before fusion, the loss on ignition (LOI) was determined by calcining the sample at 975 ºC for two hours. International standards from different geological services have been used for the elaboration of the different calibration lines. For the determination of trace elements, the Pro-trace program of the PANalytical company was used, based on calibration curves that include both geological patterns and reference patterns of the program itself. The samples were prepared in a tablet pressed at 40 tons for two minutes. The amount of sample used was 10 g mixed with a solution of Elvacite, in a proportion of 10 g of sample with 4 ml of solution.

The SoilGen model was then used to model the measured CDF values, assuming one completely inert (0-weathering rate) mineral, and to explore the sensitivity of the CDF to variations of precipitation. The effect on model results of a marked

gradient of precipitation, between 200 and 1200 mm, has been evaluated at the SC10 location, because of its position at the summit, representative of the larger plateau area.

We conducted the performance evaluation by selecting various statistical indicators to conduct a quantitative analysis: the fraction of predictions within a factor or two (FAC2), the fractional bias (FB), the root mean square error (RMSE), and the normalized mean square error (NMSE).

**Table 3. Chemical composition of the seven soil profiles (SC4-SC7). Values are averaged over the total soil profile depth. Mean measured Chemical Depletion Fraction, CDF.**

| Location | Soil profile depth (m) | $SiO_2$ | $Al_2O_3$ | $Fe_2O_3$ | MnO | MgO % | CaO | $Na_2O$ | $K_2O$ | $TiO_2$ | $P_2O_5$ | L.O.I | Zr ppm | CDF |
|----------|------------------------|---------|-----------|-----------|------|-------|------|---------|--------|---------|----------|-------|--------|------|
| **SC4** | 0.44 | 70.8 | 16.5 | 2.3 | 0.03 | 1.1 | 0.56 | 2.6 | 4.4 | 0.40 | 0.12 | 0.75 | 148.0 | 0.19 |
| **SC5** | 0.60 | 66.7 | 17.0 | 5.1 | 0.06 | 1.7 | 0.28 | 2.2 | 4.6 | 0.83 | 0.11 | 1.0 | 284.9 | 0.04 |
| **SC6** | 0.55 | 68.0 | 16.8 | 3.9 | 0.07 | 1.2 | 0.27 | 1.9 | 5.3 | 0.69 | 0.11 | 1.4 | 314.2 | 0.13 |
| **SC7** | 0.97 | 69.8 | 16.5 | 2.6 | 0.07 | 1.1 | 0.46 | 2.9 | 4.4 | 0.48 | 0.11 | 1.1 | 181.3 | 0.26 |
| **SC8** | 0.47 | 62.2 | 18.7 | 5.7 | 0.11 | 3.1 | 0.51 | 1.6 | 4.7 | 1.03 | 0.15 | 1.7 | 294.0 | 0.20 |
| **SC9** | 0.57 | 69.1 | 17.1 | 2.8 | 0.05 | 1.3 | 0.45 | 2.9 | 4.2 | 0.48 | 0.11 | 1.2 | 166.6 | 0.19 |
| **SC10** | 0.51 | 69.9 | 16.6 | 2.7 | 0.03 | 1.1 | 0.61 | 3.3 | 4.0 | 0.46 | 0.15 | 1.1 | 160.6 | 0.19 |

X-Ray Diffraction (XRD) analysis was performed along the catena to determine the bulk mineralogy. Analyses were conducted using a Bruker D8 Advance instrument using Cu Kα radiation. Samples were ground to a fine powder (~50 μm) and analyzed

at a rotational speed of 1.116 º2θ min$^{-1}$. X-rays were generated at a setting of 40 kV, 40 mA through a range of 3–70◦(2θ). Peak data for each sample were analyzed using DIFFRAC.EVA software and compared to Whittig & Allardice (1986) to determine the mineralogy of the sample.

**3 Results and discussion**

**3.1 Model testing and results**

The model was tested using in-situ soil moisture observations, in order to check that the model represents well the soil profile's hydrological balance before modelling chemical weathering reactions and material fluxes. The model was adjusted for rainfall interception by vegetation. Figure 4 shows the $R^2$ values, with the best correlation determined by the daily precipitation reduced by a fixed value of 2 mm and interception evaporation fraction of 0. Consequently, all further simulations were carried out using this combination. The results are shown in Fig. 5, which compares the daily values of simulated and measured soil moisture content across a period of 1 year and 1-month (from November-2016 to December-2017) at the SC4 location, at the summit of SFS. Soil moisture values, both measured and simulated, vary between approximately 0.05 ($m^3m^{-3}$) and 0.30 ($m^3m^{-3}$). The calibrated coefficient of determination ($R^2$) and the Nash-Sutcliffe Efficiency (NSE) index indicate a good fit of SoilGen to the observations, with values of 0.85 and 0.78, respectively. Overall, most values fall close to the 1:1 line, although there is a small overprediction of low soil moisture values and underprediction of high soil moisture values. However, given the complex soil conditions and variable Mediterranean climate, this prediction can be considered a valid representation of the actual soil hydrological dynamics.

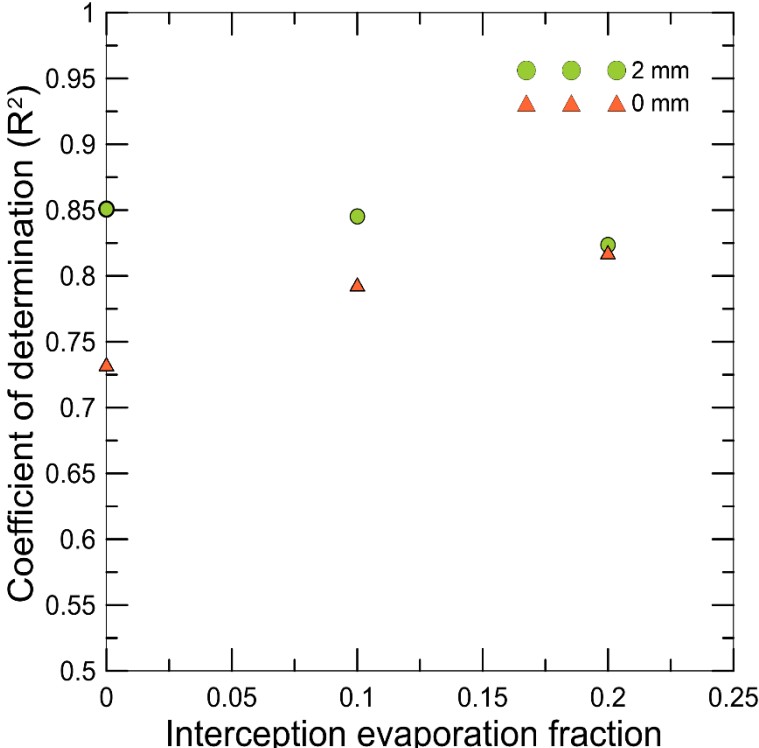

**Figure 4: Coefficient of determination ($R^2$) calculated for different Interception evaporation fractions with the daily precipitation reduced by a fixed value of 2 mm and without reduction or 0 mm.**

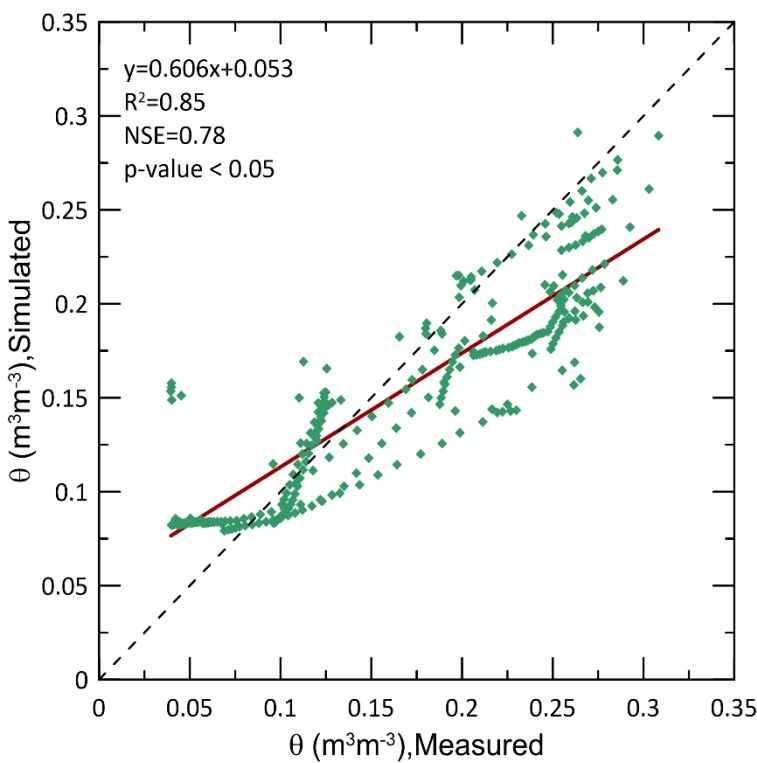

**Figure 5: Relation between soil moisture content (θ) values simulated and measured at SC4 location. The coefficient of determination ($R^2$) and the Nash-Sutcliffe Efficiency (NSE) index are indicated.**

A comparison between measured and modelled CDF values is shown in Fig. 6. Measured CDF values indicate an intermediate weathering of the soil profiles, with values ranging between 0.04 and 0.26. The highest chemical weathering was measured in the profiles SC7 and SC8, on the NFS. Simulations with SoilGen generally represent the trend in the measured data well, although the data show an overall trend for CDF to be overpredicted (Fig. 6), as the average slope of the regression line ($CDF_{measured} = 0.630 \cdot CDF_{calculated} + 0.011$) is lower than 1. The 1:1 solid line (perfect model) and the 1:0.5 and 1:2 dashed lines (FAC2) were added to the scatter plot (Fig. 6) to assist the interpretation of the results. The results of the statistical metrics that are used for the quantitative evaluation of the results met the criteria suggested by Kumar et al. (1993) and were also adopted by other authors such as Kontos et al. (2021) or Boylan & Russell (2006): the performance of a model can be deemed as acceptable if FAC2 ≥ 0.80, being the ideal value 1.0 (100%), NMSE ≤ 0.5, and -0.5 ≤ FB ≥ +0. 5, Taken this into account, the obtained FAC2, NMSE, and FB values are quite acceptable and equal to 0.86, 0.22 and -0.39, respectively. According to the FAC2 value, 86 % of simulated values were within a factor of two of the measured values. A negative value of FB indicates model overestimation whereas RMSE and NMSE do not account for over or underestimation but their ideal value is zero (Brancher et al., 2020). The model, therefore, represents the measured trend in CDF values correctly, although there exists a positive bias. However, given the fact that the model is uncalibrated in terms of chemical reactions and with respect to the resulting soil properties, we consider this result to be satisfactory. Indeed, some degree of bias can be expected,

in part since this 1-D model is not set up to handle possible lateral fluxes of water and chemical weathering products but also

because of the large time scale of the modelled processes, a common problem in this type of soil formation studies (Opolot & Finke, 2015).

The low CDF in SC5 location, considering the minimum Zr value measured in the soil profile, indicates that the complete soil profile is slightly weathered. This is associated with lower $SiO_2$ and higher MgO values with respect to the other locations (Table 3). Also the XRD results (Table 4) indicate localized differences in bedrock mineral composition at this location. The

dominant minerals in the catena are quartz, feldspars, and other phyllosilicates such as mica. Kaolinite and Chlorite-Vermiculite intergrade are accessories. However, the SC5 rock samples differ from the rest of the catena by a lower amount of Mica, lower Quartz content, and higher Chlorite-Vermiculite content which seems to support the results of the chemical composition analysis of Table 3. Further evidence corroborating the different rock composition at the SC5 site and its consideration as an outlier is given by the surface topography, as shown in Fig. 7. It can be seen that a small knickpoint is

present just upslope of SC5, which could indicate indeed a lower weatherability around SC5. Also the SC8 location is characterized by lower $SiO_2$ and high MgO values, and different mineral composition. However, the simulated CDF at this location does not deviate as much from the measured values as in SC5. This location on the NFS shows denser vegetation than location SC5 on the SFS (Fig. 7), due to the higher water availability on the NFS (García-Gamero et al., 2021). This could increase the acid supply, and promote weathering at the SC8 location. The effect of vegetation is certainly complex, as other

authors such as Oeser & von Blanckenburg (2020) pointed out that the presence of denser vegetation might counteract a potential weathering increase in their study along the climate and vegetation gradient in the Chilean Coastal Cordillera. They attribute this to an increased nutrient cycling, resulting in an absence of a correlation between CDF and mean annual precipitation in their study. Dong et al. (2019) pointed to complex ecohydrological and soil thickness feedbacks to understand weathering in karst landscapes, which they found to be highest at intermediate soil depths.

This could explain why SC8 is not an outlier, similar to SC5. While the underlying reasons for any deviations between modelled and measured CDF values are certainly complex, they will be explored below in more detail, in order to try to establish systematic relations between CDF and topographic and hydrological variables.

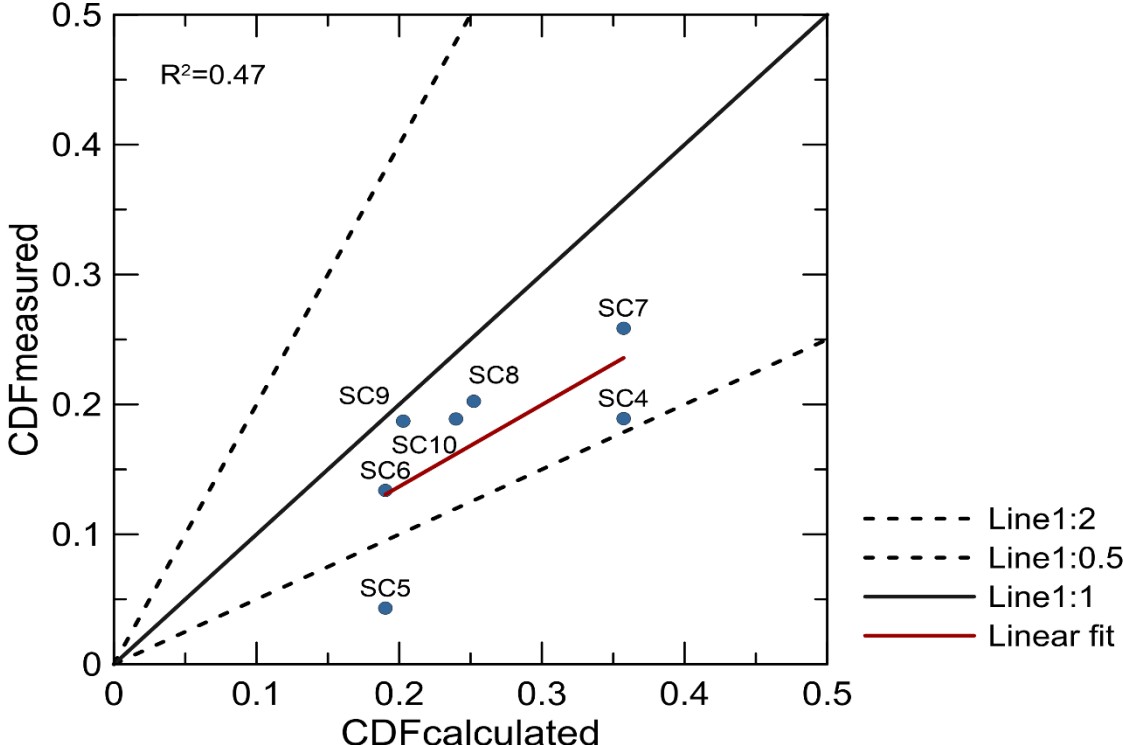

**Figure 6: Relation between CDF values calculated based on simulations and measured based on field samples. The 1:1 line (solid) represents the perfect model and the 1:0.5 and 1:2 lines (dashed) embrace the data within a factor of two (FAC2). The coefficient of determination ($R^2$) is indicated.**

.

**Table 4. Bulk mineralogy along the catena, analyzed by X-Ray Diffraction (XRD).**

| Location | Mineral Composition | | | | | |
|---|---|---|---|---|---|---|
| | Quartz | Feldspar | | Mica | Chlorite- | Kaolinite |
| | | (Plagioclase,Orthoclase) | | (Muscovite/Biotite/Illite) | Vermiculite | |
| SC4 | X | X | X | X | X | X |
| SC5 | X | X | X | (X)[a] | X | X |
| SC6 | X | X | X | X | X | X |
| SC7 | X | X | X | X | X | X |
| SC8 | X | X | X | | X | X |
| SC9 | X | X | X | X | X | X |
| SC10 | X | X | X | X | X | X |

[a] At the SC5 location, mica content was markedly lower compared to the other points, see Figure S1 in the Supplement.

### 3.2 Topographic and hydrological effect in CDF

The SoilGen model was applied to test the hypothesis that CDF values could be explained by landscape position and simulated by a simple one-dimensional model. Any deviations can then be analyzed to identify model shortcomings and future needs for model improvement.

To analyze the effect of topographic position along the catena, measured and simulated CDF values together with the conceptual model of the subsurface of the Critical Zone (the zone of the Earth surface that extends from the top of the canopy to the bottom of the groundwater) in the study site are shown in Fig. 7. As mentioned in Fig. 6, simulated and measured CDF values generally have the same trend, except for SC5. Higher simulated values were observed on the summit, SC4, and the toeslope, SC7, followed by location SC8 in the backslope of the NFS (Fig. 7). These locations were among those with the lowest local slope gradient, except for SC8.

The position along the toposequence alone did not explain well the spatial variation of the CDF, so it has to be concluded that its variation is due to other factors that are considered in the model and will be analyzed.

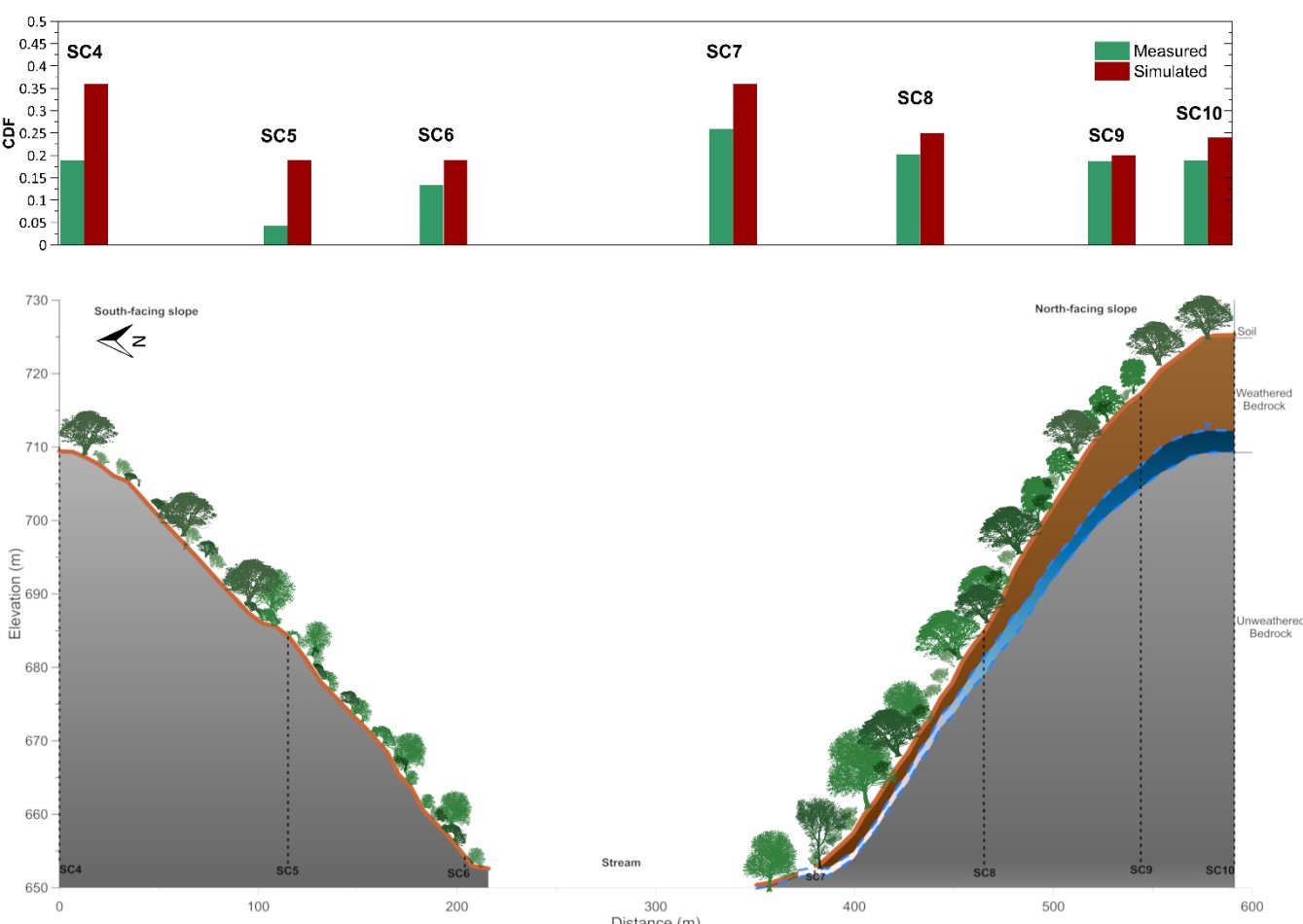


**Figure 7: Variation of measured (green bars) and simulated (red bars) chemical depletion fraction (CDF) (top panel) along the studied north-south catena, including a schematic representation (bottom panel) of the vegetation, surface topography, soil, weathered bedrock, and unweathered bedrock. On the NFS we also distinguish a seasonal groundwater table (blue).**

The relation of CDF values with different external factors was considered in Fig. 8. The variation of CDF along the

toposequence was related to local slope gradient (Fig. 8a) and because the role of soil water hydrology on chemical weathering with hydrological parameters, i.e., infiltration (I, $m^3$ $yr^{-1}$) (Fig. 8b), average moisture content ($\theta$, $m^3 m^{-3}$) (Fig. 8c), and water residence time (RT, yr) (Fig. 8d). The hydrological parameters were derived from the simulations. We also explored the relation with other variables, such as curvature and upslope contributing area (not shown in Fig. 8), but found no significant relation. This corroborates field measurements by Şensoy & Kara (2014) that have shown that the effect of curvature on runoff

production is not clear.

The results indicated an absence of correlation with the current slope (Fig 8a), although the highest CDF values were found for two soil profiles with a low slope gradient (SC4 and SC7), this was not the case for SC10. On the other hand, the data indicate a positive relationship with average moisture content and infiltration. This implies that higher infiltration leads to

higher chemical weathering. A negative relation was observed with residence time, implying that a faster throughflow of
reactive water from rainfall speeds up the weathering process. The absence of statistically significant relations for slope and
these hydrological parameters could be due to the complexity of the modeled processes, or due to the geomorphology and
landforms. One reason for this is the long-time scale of modelling, and the current topography might not have been constant
during the full period of weathering. The slope gradient is kept constant for the entire soil profile simulation time of 20 000
years as steady-state topography is assumed, following for instance Rempe and Dietrich (2014). This is a valid assumption in
slowly eroding natural landscapes, although in 20 000 years the slope gradient of the sample points could have changed. This
is a common problem in historical modelling of soil erosion in agricultural landscapes as the slope diminishes with time due
to erosion and the flattening out of the landscapes. In such agricultural landscapes, the steady-state assumption is not valid, as
erosion rates are quite high, but in any case, erosion is modelled using current topography because of a lack of a better
alternative. Some authors (Peeters et al., 2006) attempted to develop backward modelling solutions, where the topography is
iteratively rejuvenated by taking into account modelled erosion and deposition rates, but these models need detailed input data
which is not usually available, and, at present, is not available in soil formation models.

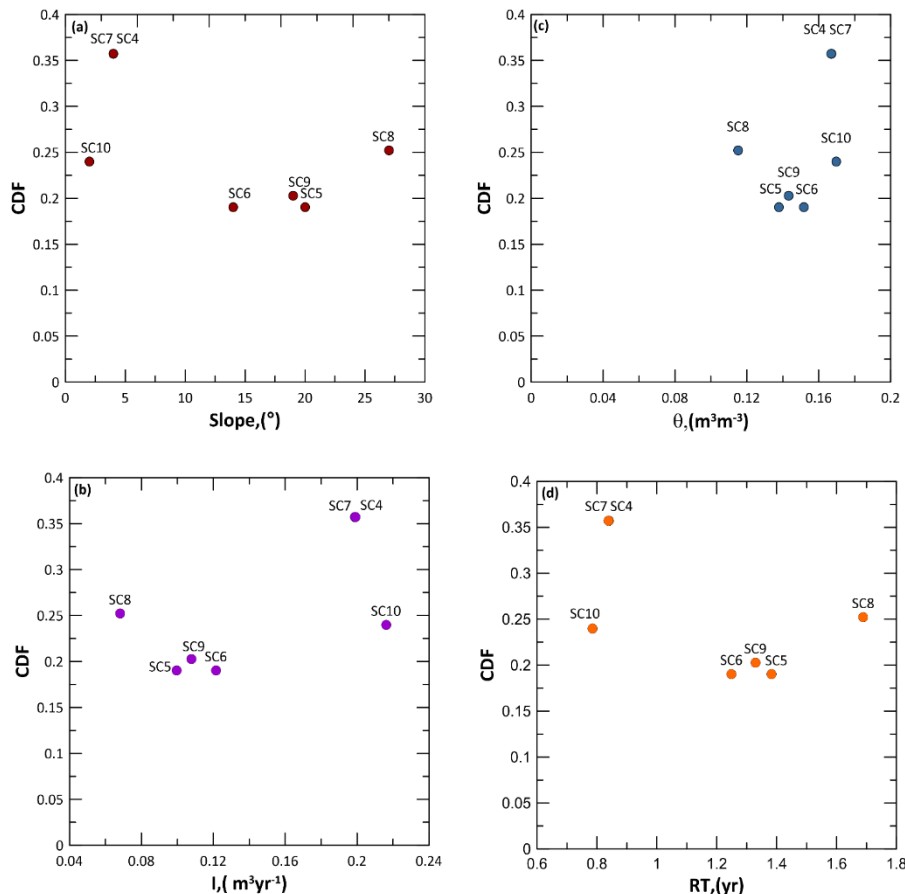

**Figure 8: Relation between CDF values and (a) Local gradient slope (°) (b) Infiltration, I (m³yr⁻¹), (c) average soil moisture, θ (m³m⁻³) and (d) Water Residence Time, RT (yr).**

The good relation between chemical weathering and hydrological variables corroborates previous studies that concluded the hydrological control on chemical weathering to be dominant (Lasaga et al., 1994; Maher, 2010; Maher and Chamberlain, 2014). It should be noted that this is well known already from early pedological research, for example, Jenny (1941) found a good correlation between carbonate precipitation and precipitation in loess soils in the US. However, the quantification of specific weathering processes is new, and the recent studies quantifying the hydrological regulation of chemical weathering processes have important implications beyond soil formation, for example on the geological carbon cycle (Maher and Chamberlain, 2014). Maher (2010) indicates that natural systems can generally be considered to be transport-controlled, where the reaction rate of chemical weathering processes depends mostly on the departure from the equilibrium. This departure is controlled by the water flow rate through the profile and how fast or slow weathered products are exported from the soil profile. In other words, in transport-controlled weathering systems, flow rates and solubility by definition will be the dominant control on mass removal. As natural systems are difficult to characterize fully, not much field data has been collected corroborating this so far. Some studies, like Schoonejans et al. (2016) found a positive relation between chemical weathering and rainfall

along a climatic gradient in southern Spain. However, the results of this study, both measured and modelled CDF values, are among the first to find differences in chemical weathering in different landscape positions along a catena. In this study area,

rainfall is homogeneous, but the differences in infiltration can be purely attributed to differences in exposition and landscape position. Further studies will be needed, not only at the landscape or regional scale but also at a shorter spatial scale, such as this catena-scale study to elucidate these hydrological effects in more detail. Some authors have pointed to important feedbacks with plants, that definitely can have a significant impact on soil hydrology and perhaps also a direct influence on chemical weathering processes (Cipolla et al., 2021; Porada et al., 2016).

The results of this study are in contrast with observations by Molina et al. (2019) on 10 toposequences in a High Andean catchment. They observed only a marginally significant topographic control on chemical weathering extent, while our data varies considerably between landscape positions. Rather than topography, Molina et al. (2019) concluded that vegetation exerted an important control, as they found highly significant differences in chemical weathering extent between vegetation communities. In their study, however, they compared very different vegetation types, ranging from forest to grass and cushion-

forming plants. In our study, vegetation is more similar between SFS and NFS, although vegetation is denser on the latter. This could have an effect on acid supply, but also on hydrology. Although counterintuitively, in previous work in the study site to characterize the hydrological dynamics using soil moisture sensors and piezometers, García-Gamero et al. (2021) observed very similar surface hydrological dynamics between the NFS and SFS. In fact, daily soil moisture storage change differences between both opposing slopes did not suggest more interception on the NFS (García-Gamero et al., 2021), despite

the denser vegetation. The denser vegetation cover on the NFS only caused it to dry out somewhat earlier during the year compared to the SFS. However, subsurface water dynamics were found to be significantly different, with a deeper weathered bedrock on the NFS (Fig. 7) that allows seasonal water storage and a significant lateral water flow. A good correlation between the normalized difference vegetation index (NDVI) and the water table evolution was found on the NFS (García-Gamero et al., 2021).

The model SoilGen could be further improved by taking into account lateral fluxes of water and sediment. The one-dimensional SoilGen model does not consider lateral fluxes by definition. These lateral water fluxes on the NFS could explain the higher measured CDF values in SC9, SC8 and particularly in SC7. The points located along this NFS all receive an additional lateral water influx, which is greatest for SC7 at the toeslope. This lateral water flux can accelerate the chemical weathering and is not considered in the one-dimensional model. At present, as far as the authors know, no soil profile formation model exists

that has this capability. On the other hand, on the SFS soils are shallower and this lateral connectivity does not exist. The other profile on the SFS, SC6 however does behave as expected and is characterized by the lowest CDF values, except for SC5 above mentioned, both measured and simulated. To take into account these lateral fluxes goes beyond the objectives of this paper that aims to model chemical weathering with a simple model, but future modelling efforts should be pointed in this direction.


### 3.3 Climatic effect on chemical weathering: Sensitivity Analysis

After analyzing the variability in soil formation along the north-south oriented catena, the effect of a simulated precipitation gradient (200-1200 mm) on the CDF was studied, at one fixed, representative location, SC10 (Fig. 9).

The lowest CDF values throughout the profile correspond to the lowest precipitation value of 200 mm, as expected. For this precipitation value, there is very little difference of CDF with depth. Increasing precipitation to 400 mm leads to a sudden increase in the weathering of the top 40 cm of the profile, with a marked depth gradient and similar CDF values in depth compared to the 200 mm case. A precipitation increase to 600 mm does not change the CDF-depth pattern much, although the weathering front lowers. In the situation with low precipitation (P=200-600 mm yr$^{-1}$), the rainfall is lower than the potential

evapotranspiration, and as a consequence, only the upper 45 cm of the profile are wetted while the lower part remains dry. This means that the contact time of meteoric water with the topsoil is long while it is near 0 in the subsoil. A long contact time allows for more weathering and this is reflected by higher CDF in the topsoil than in the subsoil. This contrast increases with precipitation increasing from 200 to 600 mm, above-mentioned. At P=800 mm yr$^{-1}$ the weathering and CDF values in depth increase markedly, the lower part of the profile also is moistened and the contrast decreases, while the contact time is still

large, resulting in a high CDF. These simulated depth patterns are comparable to those measured in the eastern part of the Betic Cordillera, where precipitation ranges between 275-425 mm, located in Southeast Spain by Schoonejans et al. (2016). However, at even higher precipitation, P > 800 mm yr$^{-1}$, the subsoil is moistened and this increases the hydraulic conductivity and thus decreases the contact time. As a result, the CDF decrease and also the topsoil-subsoil contrast in CDF disappears. (Fig. 9). The latter simulated depth patterns are comparable to those measured by Oeser & von Blanckenburg (2020) in their

study, above mentioned, in an arid to humid climate and vegetation gradient in the Chilean Coastal Cordillera. These authors found that the average CDF for the profile (combined analysis of NFS and SFS profiles) amounts to 0.25 in the humid-temperate site, in granitic rock (granodiorite) with a mean annual precipitation of 1084 mm. This value is similar to that simulated for precipitation of 1000 mm in our work (Fig. 9).

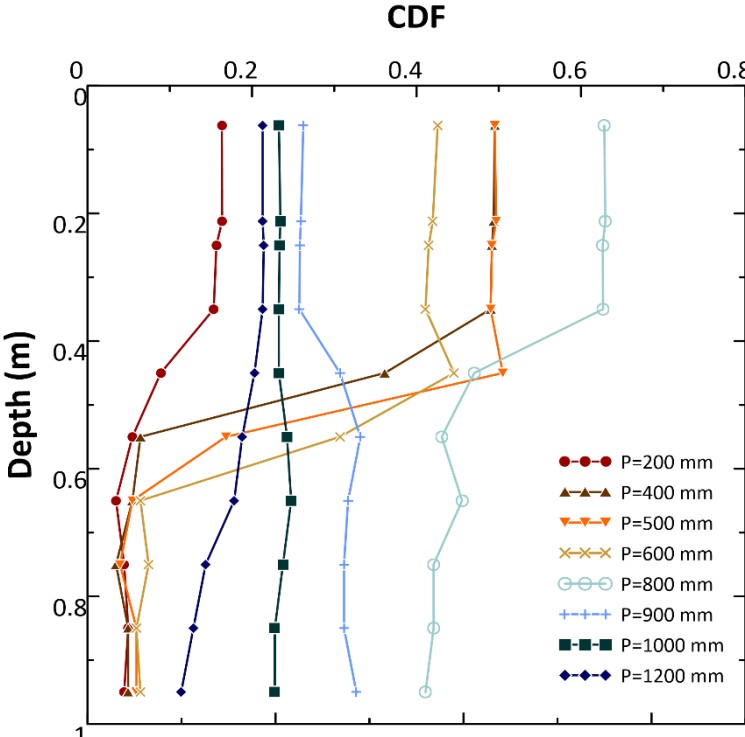

 **Figure 9: Weathering intensity (expressed by the chemical depletion fraction, CDF) of a soil profile under different annual rainfall (P). Profiles are simulated in the SC10 location, at the summit.**

Figure 10 then shows the total CDF, averaged over the entire soil depth, i.e., 1 m of the soil profile. The results from this study, represented by the blue dots, showed an increased weathering up to 800 mm, but after this critical value, the profiles were less weathered. Moreover, in Fig. 10 CDFs of this study are compared to a global data compilation of granitic soil-mantled

hillslopes (Schaller & Ehlers, 2021). Also, this global dataset seems to indicate that high weathering rates measured as CDF are not obtained with the highest precipitation values, but rather with intermediate precipitation. With precipitation less than 800 mm, a substantial part of the year will have a precipitation deficit. This means that leaching is of less importance and concentrations of cations released by weathering in the soil solution will be high. One theory often applied to quantify weathering is the transition state theory (TST). In the quantification of weathering fluxes by TST, the dissolution rate of a

mineral is the product of the rate parameter (which is often a function of pH) and the degree of saturation of the soil solution with respect to the dissolving mineral. Drier climates often result in a higher pH and a higher degree of saturation, which both (and certainly in combination) lead to lower weathering rates for most minerals. The decreasing weathering rates imply decreasing CDF values with lower precipitation. The lower the precipitation, the higher the deficit, leading to a high CDF in only that part of the soil that becomes moistened, because the contact time is large (no leaching). The lower precipitation, the

thinner the part that is moistened, and thus the lower the profile-average CDF. At higher precipitation up till 800 mm, the

moistened part of the soil profile becomes thicker and thus increases the profile-average CDF. This is the upward branch of the Albrecht curve.

At still higher precipitation, the whole soil becomes moist, with higher hydraulic conductivity and increasing water flow velocities throughout, hence lower contact time and decreasing CDF. Thus, the downward branch of the curve represents a situation of increased leaching but at the same time, less intensive weathering because the contact time decreases.

Additionally, vegetation will be lusher, and then the nutrient pump may prevent leaching of some elements. Both will decrease the CDF.

This threshold or maximum weathering correspondent to intermediate precipitation is coincident with the maximum generalization of Albrecht's curve, which is shown in Fig. 10 by the grey shaded area. Huston (2012) pointed out the link between climate and soil formation and properties that scientists such as William Albrecht generalized 80 years ago. Albrecht's curve (e.g. Albrecht, 1957) is a rule that illustrated the effect of precipitation on soil-forming processes and soil properties (Huston, 2012). In this diagram, which presents a maximum in the center, the effect of a precipitation gradient on the rates of physical, chemical, and biological processes that affect pedogenesis and important soil properties are shown. Different later works that have studied soil properties and weathering along a marked precipitation gradient (e.g. Chadwick et al., 2003) confirmed Albrecht's curve. Similarly, Dixon et al. (2016) in their study for chemical weathering in postglacial soils of New Zealand found an important pedogenic threshold coincident at mean annual precipitation of ~800 mm yr$^{-1}$, very similar to the threshold value that was identified in this sensitivity analysis. Oeser & von Blanckenburg (2020) on the other hand, in their study in the EarthShape Critical Zone located along the Chilean Coastal Cordillera, found no correlation between the degree of weathering and mean annual precipitation. Therefore, they pointed out that a competitive effect seems to offset the expected increase in the rate of weathering with precipitation. Huston (2012) even related this concept of the Albrecht curve to ecosystem trends. He analyzed global variations in soil properties, Net Primary Productivity and biodiversity as a function of precipitation. He found similar, unimodal curves with maximum values for soil properties such as total exchangeable bases and found that the maximum corresponded to the point where precipitation is equal to the evapotranspiration rate. Given the importance of soil properties and chemical weathering of soil profiles for ecosystem response, these results can be far-fetching consequences and should be explored further with more simulations and profile data.

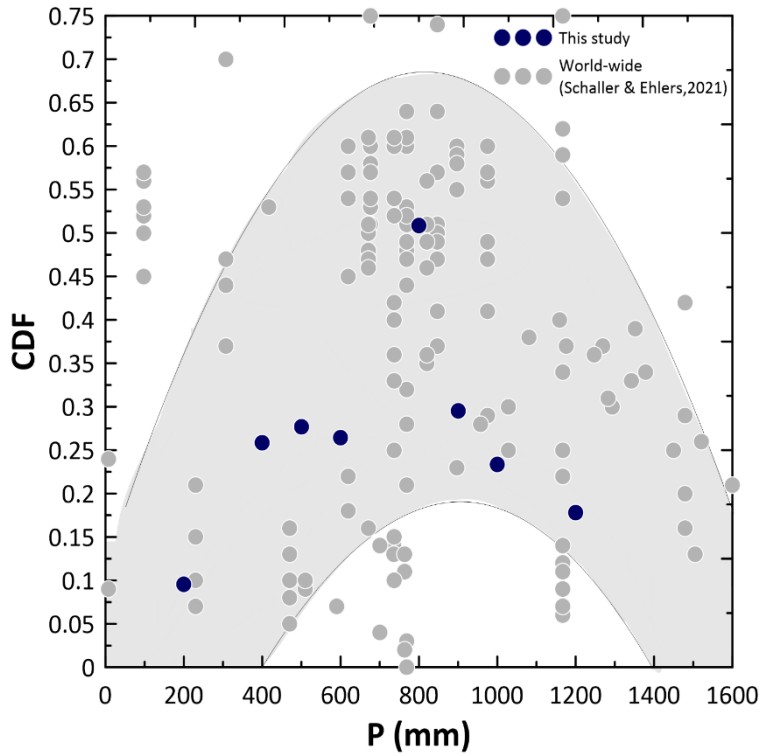

**Figure 10: Chemical weathering (CDF) versus annual precipitation. The expected trend is indicated by the shaded area and follows a parabolic pattern with maximum chemical weathering for intermediate precipitation. Observations from this study are**
**depth-averaged (blue dots). Observations from other study sites situated in granitic soil-mantled hillslopes (grey dots) are from a review by Schaller & Ehlers (2021).**

## 4 Conclusions

The effect of topographic position and hydrology on chemical weathering was analyzed in soils formed on granites under a Mediterranean climate. Chemical depletion fraction was measured and modelled in seven locations, selected along a north-
south oriented catena in southern Spain. The model SoilGen was used to simulate pedogenesis and the measured chemical weathering status over a 20000-year period.

Despite the complexity of the catena and the hydrological conditions, a good correspondence was obtained between modelled and measured CDF. The variability of CDF values was explained better by hydrological variables than by topography. No clear relation to the catena position was found. No relation with slope gradient was observed. However, the CDF data did
indicate a positive relation to the hydrological variables of soil moisture and infiltration, and a negative relation to water residence time. In addition, differences between measured and simulated CDF values could be attributed to lateral water fluxes that are not considered in the model.

The model sensitivity was evaluated with different precipitation regimes. The results showed a marked depth gradient for rainfall under 800 mm for the CDF, but it showed a uniform depth distribution for precipitation above 800 mm. For the profile
average chemical weathering, maximum values were observed for intermediate precipitation values, around 800 mm.

**Data availability**

Data are available upon request to the authors.

**Supplement**

**Author contribution**

TV and VGG conceived and designed the study with substantial input from PF and AP. ARS and TV performed the sampling. The data curation and formal analysis, methodology, and visualization for the paper were performed by VGG, with substantial input from PF and TV. VGG and PF performed the simulations. VGG wrote the first draft. All the authors (VGG, TV, AP, ARS, and PK) contributed to generating and reviewing the subsequent versions of the paper.

**Competing interests**

Peter Finke is topical editor of SOIL journal. All other authors declare that they have no conflict of interest.

**Acknowledgments**

This work is supported by the project "Estableciendo un Observatorio de la Zona Crítica para la Hidropedología y Agricultura Sostenible en el Mediterráneo" (AGL2015-65036-C3-2-R) funded by Programa Estatal de Investigación, Desarrollo e Innovación orientada a los retos de la sociedad 81/150 para el cuatrienio 2016-2020 (MINECO/FEDER, UE). Vanesa García-
Gamero was awarded with a FPU fellowship (FPU15/05279) from the Spanish Ministry of Education, Culture and Sport. This paper was the result of a research stay of Vanesa García-Gamero at Ghent University, funded by the fellowship "Becas Movilidad Internacional" (2017-2018) from Universidad de Córdoba.

Vanwalleghem thanks funding to the Department of Agronomy by the Spanish Ministry of Science and Innovation, the Spanish State Research Agency, through the Severo Ochoa and María de Maeztu Program for Centers and Units of Excellence in R&D
(Ref. CEX2019-000968-M).

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
