# Peer review of "Modelling the effect of catena position and hydrology on soil chemical weathering"

_SOIL, 2021_

## Referee Comment (RC2)

[referee-annotated manuscript omitted]

---

## Author Comment (AC1)

**Topical Editor initial decision: Start review and discussion**
by Jeffrey Homburg
**Comments to the author**:
This paper presents an important advance based on the Soil-Gen model to integrate hydrological properties, both measured and simulated, with topographic-climatic data to explain variability in soil chemical weathering for a catena. Based on a number of measures, the 1-D model performed surprisingly well in predicting variability and a threshold in weathering intensity. I have a number of questions, but most of those can wait until the referee comments are completed. One question I had was about concave-convex combinations across and down slopes for the specific sampling locations (because those properties can vary over short distances on slopes and they can have strongly affect the amount of water that infiltrates at a single point. (As a side note, I found it interesting that you reference the Albrecht curve, because Albrecht of Missouri trained the pedologist (Bob Miller at Lousiana State University, an expert in clay/soil mineralogy, loess, and Quaternary geology) who first trained me in soils in grad school in the 1980s!)

**We thank the editor and reviewers for their positive evaluation of the manuscript. We appreciate all of the valuable comments from the reviewers of our work.**

**We are particularly happy that our work that supports the Albrecht curve was of interest. We agree it is an interesting concept that merits "rediscovery" by the soil science community.**

**We have revised our manuscript, according to the reviewers' comments, questions, and suggestions. In what follows the reviewers' comments are in** black **and the authors' responses are in blue.**

**Regarding concave-convex combinations, we have calculated profile and plan curvature and added them to Table 1, together with a short discussion. However, after revising the literature on this, we found that, while from a theoretic standpoint the influence of curvature on runoff and soil moisture is clear, field evidence is unclear on whether curvature (concave, convex, or uniform) has such a marked effect on runoff. Under field conditions, we found that observations do not always support that curvature has a strong effect on runoff and thus on soil moisture (for example, Sensoy and Kara, 2014). Their field measurements of runoff did not distinguish statistically between concave-convex slopes (although uniform slopes did produce significantly higher runoff compared to the concave-convex ones). We believe that in our setting with coarse-textured soils and a high, natural vegetation cover, its influence will also be limited, although, as commented, we included the data and a short discussion.**

RC1: 'Comment on soil-2021-78', Anonymous Referee #1, 08 Nov 2021

After reading this manuscript, I question whether the term "catena" is the proper description here. A true catena will have differences in parent materials, vegetation, etc. across the catena. This location appear to have largely uniform parent materials and vegetation. I believe "toposequence" would be a more accurate depiction of the study site than "catena".

**Answer:**
**We do not agree with the reviewer on this point. The concept "catena" formalized in the 1930s by Geoffrey Milne does not necessarily imply a difference in parent material (Borden et al., 2020). A catena implies a mass change along with the different profiles that comprise the catena. In other words, there needs to be a landscape relation between the soil profiles and a transfer of mass and energy, which can be through erosion-deposition or lateral transfer of water and solutes. See for example the definition by Hall (1983) in Developments in Soil Science: "*The concept of catena has been modified (Bushnell, 1942) and is now used almost interchangeably with toposequence by many, particularly in the United States. Toposequence presently carries with it a morphologic connotation; a change in colors, predominantly a change in grayness, that is related to relative elevation and thus to changes in hydrology. Catena on the***

*other hand carries with it a process-response connotation. The soils of a catena differ not only in morphology but are considered to differ as a result of erosion, transport, and deposition of surficial material as well as leaching, translocation, and deposition of chemical and particulate constituents in the soil."*

Line 27 – The 5 soil forming factors should not be capitalized.

**Answer:**
**We have corrected the capitalization.**

Lines 74-77 – It is nice that Brantley et al. (2017) developed a conceptual model in three different parent materials, but what is the application/advantage of this model. Why is it important to bring this model up in the introduction? More to the point, what did they conclude as the relationship between reaction fronts and catena position?

**Answer:**
**Firstly, please note that the reference to the Brantley paper was mistaken, perhaps this created this question, for which we apologize. The correct reference is Brantley et al., 2017. Toward a conceptual model relating chemical reaction fronts to water flow paths in hills. Geomorphology 277:100-117.**

**This paper by Brantley et al. (2017) is important here because their work illustrates very well the increase of factors that one has to consider in order to understand weathering differences along a hillslope, i.e. that one has to look further than precipitation and evapotranspiration, also important are lithological considerations, lateral flow, specific flow paths that are influenced by veins, fractures and faults, and erosion-deposition for example.**

**Concerning conclusions on reaction fronts and catena position, they consider the entire hillslope, so it is difficult to translate exactly to catena positions, but for example, for the ridgetop position they make clear statements and found important differences in weathering depth depending on lithology. The variation of weathering depth along the hillslope is illustrated in their figure 5:**

[Figure]

However, their main conclusion from their conceptual model is possibly that it is important to take into account the presence of two main zones of water flow, interflow and a groundwater zone. The depth intervals of water table fluctuation for interflow and groundwater flow are reaction fronts characterized by changes in composition, fracture density, porosity, and permeability. We expanded the discussion in the text to reflect these main conclusions. However, the authors also admit more work and data will be needed to translate this into a numerical model or quantifiable data for different catena positions.

Line 109 – The introduction gives a rather abrupt transition from discussing previous works to presenting the objectives of this study. A better transition would be beneficial. And, the objectives should be in a paragraph of their own.

Answer:
We have reworked the introduction to make this transition better and separated the objectives as suggested.

Lines 132-133 – The scientific names of the most common vegetation species should be supplied here.

Answer:
We have included the scientific names of the most common vegetation species. The new sentence reads as follows "Vegetation in a dehesa includes sparse trees, holm (Quercus Ilex L.) and cork oaks (Quercus suber), shrubs, retama, and annual grasses such as Lolium sp., Bromus sp., and Trifolium sp., with

**maximum production in spring and a non-vegetative period in summer (Olea and San Miguel-Ayanz, 2006)".**

Table 1 – The latitude in this table isn't meaningful, because it isn't different for any of the profiles. Just say the study site is at about 38 degrees N in the site description in materials and methods. Same comment with the downwind bearing.

**Answer:**

**We thank the reviewer for the suggestion. We have mentioned the latitude and the direction of the dominant winds in the study site description only and removed them from Table 1.**

Figure 2 – It appears that climate and vegetation were varied from 12,000 years BP to present, but were considered constant from 20,000 BP to 12,000 BP. There are not any studies from this area that could provide climate and vegetation data for the 20,000 to 12,000 BP period? This is a definite weakness of the study.

**Answer:**
**The climate reconstruction is based on the work of Davis et al.,2003, which is available for the last 12,000 years, and updated by Mauri et al. (2015, Quaternary Science Reviews). To the best of our knowledge, there are no longer records available at the resolution needed for the model and that includes the climate variables required in this model (for example Jiménez de Cisneros and Caballero, 2013. Natural Science, that only has temperature based on speleothems). Locally, some longer climate reconstructions are available but not representative to our area, or without quantitative climate reconstruction data (for example Jiménez-Moreno et al., 2019. Global and Planetary Change)**

**In any case, a previous study by Keyvanshokouhi et al. (2016, Science of the Total Environment) analyzed how uncertainty in the boundary conditions affected the results of the SoilGen model and concluded that climate data uncertainty at the beginning of the series did not affect results significantly. Thus, our climate data uncertainty between 20.000-12.000 years BP is (1) the best available data we could find, and (2) we are confident a possible error here would not have a significant impact on the results.**

Lines 180-185 – Were any known samples analyzed as a quality check on the data generated?

**Answer:**
**We are not sure what the reviewer refers to exactly. We analyzed reference patterns with known concentrations. We have clarified this in the revised text.**

Table 3 shows chemical composition values for the profiles. However, with the exception of Zr, there is no explanation regarding where this data came from. This needs to be supplied. If the data was generated as a part of this study, how it was generated needs to be explained in the materials and methods. If it was generated as part of a previous published study, that needs to be referenced.

**Answer:**
**All data from Table 3 were analyzed in this study and not published previously. We further elaborated on this in the material and methods section and we hope that this solves the referee's doubts.**

**Revised text:**

**"The determination of the major elements was carried out from a fusion bead, with Lithium tetraborate and a flux/sample ratio of 10:1. Before fusion, the loss on ignition (LOI) was determined by calcination of the sample at 975 °C for two hours. International standards from different geological services have been**

**used for the elaboration of the different calibration lines. For the determination of trace elements, the Pro-trace program of the PANalytical company was used, based on calibration curves that include both geological patterns and reference patterns of the program itself. The samples were prepared in a tablet pressed at 40 tons for two minutes. The amount of sample used was 10 g mixed with a solution of Elvacite, in a proportion of 10 g of sample with 4 ml of solution."**

Line 220 – What statistical technique was used to determine that the slope of the line was significantly different than 1?

**Answer:**
**We apologize for the wrong choice of this word we did not mean significantly in a statistical sense, we rather meant "it can be seen clearly how the slope differs from 1…". To clarify this, we have deleted this sentence.**

Table 4 is not needed. Almost all of these values were already given in the manuscript in line 224. Tables should not simply repeat what is in the manuscript (or vice versa).

**Answer:**
**The reviewer is right. We have removed Table 4.**

Line 264 – There are established models with defined topographic positions published in the literature. I do not recognize the model you are using, please provide a reference for it and briefly describe it in the materials and methods. If you are not using an established model, you should either 1) use an established model, or 2) give a complete explanation of your topographic position model in the materials and methods along with an explanation as to why you are using it and not an established, published model.

**Answer:**
**We are not sure what the reviewer refers with the topographic model:**

**(1) if the reviewer refers to models such as the TPI (Topographic Position Index), we do not think those kind of models are helpful here. With the comment on line 264, we mean that we want to show where each profile is located along the catena and show the results of CDF. We just want to show a transect of the surface topography, not make a standardized classification of the landforms.**

**(2) If the reviewer refers to the model shown in the figure below, from Dixon (2015, Developments in Earth Surface Processes) that was based on Schaetzl's famous work on Soils: genesis and geomorphology (2013), then we agree we should change our terminology. That is in fact what we meant.**

**In order to standardize, we replaced hilltop by summit; mid-slope by backslope, and valley-bottom by toeslope and upper-slope by shoulder. As suggested, we have added the description of the topographic position of each profile to table 1.**

[Figure]

Lines 318-320 – Is it possible that the deeper weathering profile on the north facing slope is because there is less evapotranspiration, leading to either 1) more water for chemical weathering to take place, 2) more water to allow for vegetative growth, thus affecting chemical weathering, or 3) both 1 and 2?

**Answer:**

**In a previous paper, we analyze the hillslope hydrology in detail. With the help of a soil moisture sensor network, we found very little difference between soil water dynamics on south vs north slopes, while our original hypothesis was that soil moisture content on south-facing slopes would be lower. Our data, therefore, do not show lower evapotranspiration on the north-facing slope. This could be attributed to higher transpiration on north-facing slopes, due to the denser vegetation, despite higher radiation on the south-facing slope. Our data also showed a good correlation between vegetation greenness (NDVI) on north-facing slopes and aquifer dynamics, indicating the contribution of subsurface water to the plants.**

**So, in conclusion, we don't think our data support 1), while we agree with 2) although the water source is most probably subsurface influx.**

Line 340 – Are the rainfall amounts in the Betic Cordillera similar to those simulated here? If so, in what way (e.g., are they similar to the mid-level rainfall amounts modelled, the higher rainfall amounts modelled, etc.)? Are there other places that can fill in the missing rainfall amounts (e.g., if the Betic Cordillera amounts are similar to the greatest rainfall levels modelled, are there other studies that can fill in the intermediate rainfall levels modelled)? Right now this modelling in section 3.3 seems weak, in that there is little to no validation. Figure 9 for CDF provides the type of information I would like to see to validate the modeling shown in Figure 8.

**Answer:**

**The average annual precipitation in the Betic Cordillera is between 275 and 425 mm (Schoonejans et al., 2016). Therefore, the range is similar to the lower rainfall amounts modelled. We plan to continue working on this issue of relating CDF to rainfall in future studies, as we believe that to present a detailed validation of the CDF-depth profiles (as shown in figure 8) for each of the modelled rainfall amounts, this would require an in-depth, case-by-case study which exceeds the objectives of this study. However, as the**

**reviewer points out as well, we did find a good amount of data at the profile-averaged level, which is summarized in figure 9 and we believe that this backs up this modelling of section 3.3.**

RC2: 'Comment on soil-2021-78', Anonymous Referee #2, 08 Nov 2021

The manuscript presents a weathering simulation of soils from Spain. The manuscript's introduction does not place well the current work in the historical...why is this study needed? How does this study fill gaps in the previous work.

Pedon data and profile morphology must be included. In addition, it is not clear what is the genesis of these soils with respect to site geomorphic history, dust or loess deposition. Micro site topography around pedons is not clear. See examples of 9-component slope models, curvature analysis that could enhance your interpretation.

No information is presented on site surface characteristics, percent cover, types of vegetation cover, soil surface integrity, degree of site destabilization due to use (is the area overgrazed?).

**Answer:**
**We have included pedon and profile morphology data as suggested. We have also included curvature data. Dust or loess deposition is not significant at this location. We assume that the landscape is in steady-state, following for example Rempe and Dietrich (2014, PNAS), and the modelled time period corresponds to the soil residence time that was derived from OSL measurements of grain burial age in a previous study (Román-Sánchez et al., 2019). We have expanded the explanation on this in the text. With respect to surface characteristics, vegetation is very similar in all locations with a soil percent cover of 75 to 100 %. In any case, this information corresponds to the present time. Simulations were carried out for 20000 years period and changes in vegetation cover enter into the dynamics. Soil surface integrity and degree of site destabilization due to use is not applicable because is a seminatural area areas as grazing cattle avoids the area of the studied transect because of its steep slopes and keeps to the flat areas that are covered with grass.**

To make the case for weathering I believe mineralogy data is an absolute requirement. Simply referring to mafic and concluding weathering when there is a difference in chemistry is not strong enough. I imagine dust influence on these pedons is substantial and CDF might not be the best approach because of this.
**Answer:**
**With respect to mineralogy, we agree and performed additional XRD analysis (Whittig and Allardice 1986) that we will add to the paper. We summarize the main results here. We also consulted Carracedo et al. (2009) who described the mineralogical composition of the Los Pedroches batholith. Our new results show that the dominant minerals are quartz, feldspars, and other phyllosilicates such as mica. Kaolinite and Chlorite-Vermiculite intergrade are accessories. In the figure the diffractograms that will be included as supplementary material of the paper are shown:**

[Figure]

X-ray patterns of the powdered rock samples. C-V: Chlorite-Vermiculite intergrade; F: Feldspar (Orthoclase, Plagioclase); K: Kaolinite; M: Mica; Q: Quartz.

**Both SC8 and SC5 rock samples outstand for the lower amount or lack of Mica, lower Quartz content, and higher Chlorite-Vermiculite content which seems to support the results of the chemical composition analysis of Table 3.**

**With respect to the dust influence, we do not think the effect is very dramatic at this site. Dust deposition rates are not very significant as compared to other parts of the Mediterranean, such as Israel. Even in the presence of some low dust deposition, the main effect will be similar in all points, so acting as a blanket over the slope, thus damping out to some extent differences that might arise from differentiated weathering due to landscape position.**

**A summary table as follows will be added to the revised text with respect to mineralogy:**

| Location | Mineral Composition | | | | | |
|---|---|---|---|---|---|---|
| | Quartz | Feldspar (Plagioclase,Orthoclase) | | Mica (Muscovite/Biotite/Illite) | Chlorite-Vermiculite | Kaolinite |
| SC10 | X | X | X | X | X | X |
| SC9 | X | X | X | X | X | X |
| SC8 | X | X | X | | X | X |
| SC7 | X | X | X | X | X | X |
| SC6 | X | X | X | X | X | X |
| SC5 | X | X | X | X | X | X |
| SC4 | X | X | X | X | X | X |

More needs to be considered during results interpretation in terms of the effect of site geomorphology and subsurface horizon characteristics.

**Answer:**
**We have expanded on this, following the changes proposed.**

I have included an annotated pdf.

**Answer:**
**The comments of the annotated pdf were all taken into account, following the reviewer's suggestions**

Comments in the annotated pdf:

**Abstract**

I suggest a complete re-write.
**Answer:**
**We have rewritten the abstract.**

Lines 15-16: Weak usage
**Answer:**
**We have rewritten this sentence**
Lines 16-18:This sentence does not tell us much.

**Answer:**
**We have rewritten this sentence**

Line 18: Not a clear statement hanging alone...merge with previous into a clear sentence.

**Answer:**
**We have rewritten this sentence**

Lines 18-19:implying what?

**Answer:**
**Slope or landscape position are often used for prediction purposed in prediction via geostatistical (regression kriging) or Artificial Intelligence (random forest, etc.) techniques. These results suggest that it is important to model soil formation mechanistically and that variables related or derived from hydrological considerations could be good predictors in the absence of data to do a full-scale modelling of soil formation.**

Lines 20-24:These are independent statements...what is your story

**Answer:**

**These are the main conclusions of our work, but we tried to rewrite to make it more coherent, sticking at the same time to the word limit the abstract requires.**

**The current abstract reads as follows:**

**The sensitivity of chemical weathering to climatic and erosional forcing is well established at regional scales. However, soil formation is known to vary strongly along catenas where topography, hydrology, and vegetation cause differences in soil properties and possibly chemical weathering. This study applies the SoilGen model to evaluate the link between topographic position and hydrology with the chemical weathering of soil profiles on a north-south catena in southern Spain.**

**We simulated soil formation in seven selected locations over a 20000-year period and compared against field measurements. Good model performance was obtained comparing modelled chemical depletion fraction (CDF) against measured CDF ($R^2$=0.47). An important variation in CDF values along the catena was observed that is better explained by the hydrological variables than by the position along the catena alone or by the slope gradient. A positive trend between CDF data and soil moisture and infiltration and a negative trend with water residence time was found. This implies that these hydrological variables are good predictors of the variability of chemical weathering.**

**The model sensitivity was evaluated with a large precipitation gradient (200-1200 mm $yr^{-1}$). The model results show an increase of chemical weathering of the profiles up to a mean annual precipitation value of 800 mm $yr^{-1}$, after which it drops again. A marked depth gradient was obtained for CDF up to 800 mm $yr^{-1}$, and a uniform depth distribution was obtained with precipitation above this threshold. This threshold reflects a change in behaviour where the higher soil moisture and infiltration lead to shorter water transit times and decreased weathering. Interestingly, this corroborates similar findings on the relation of other soil properties to precipitation and should be explored in further research.**

**Introduction**

**Answer:**

**We have rewritten and restructured the introduction section, as suggested by the referee, we have reorganized the studies in chronological order and clarified the objectives. All other comments (from line 30 to line 109) in this session have been taken into account following the referee's suggestions.**

Line 30:temporal?

Lines 44-45:what defines success?

Line 61: affect what

Line 67:the importance

Line 67:I do not think kinetics is the right choice of word for the processes, no?

Lines 68-71:climate drives temp and moisture chnages....hydrology is driven by climate and site (geomorph conditions). These few sentences in this area are too vague and unspecific.

Lines 72-73:Poor topic sentence...the introduction needs to be better organized too.

To be clearer on what is a field study and what is modeled, or both, I suggest presenting the studies in chronologic order. The ideas in this paragraph are not well linked either...what is the point of speaking

about all of these? How do they relate to your study and what are the next steps from these studies that your study addresses?

Line 91: Merge the sentences...do not start sentences with 'they'..it is unspecifc who you are referring to.

Line 109:The specific objectives of your study??

How do all these other studies relate to yours? Why is your study needed or how does it move our science forward in relation to this long list of other studies you just presented?

**Material and Methods**

Line 119: all profiles went to bedrock?
**Answer:**
**Yes, all profiles went to saprolite or weathered bedrock. The main difference is that on the SFS it was observed that this saprolite is quite hard and probably very little weathered and comparable to unweathered, hard bedrock while on the NFS we observed the presence of highly weathered saprolite below the soil profile. An additional study that escapes the scope of this paper showed that hard bedrock there was found at 9.50 to 18.20 m depth.**

Line 120: By horizon or just depth...if by depth you missed the key pedogenic development pathways...depths are just about never consistent in pedogenic development across a hillslope.

**Answer:**

**The sensors were installed at fixed depths. This is standard practice in hydrological studies (Salve et al., 2012).**

**We calibrated the SoilGen model against the averaged soil moisture in the profile. Therefore, we believe that horizon distribution does not have a big impact here, as there are no horizons that act as boundaries to water flow in this profile (such for example a compacted Bt horizon).**

**In addition, the initial conditions for all soil formation simulations were homogeneous, so this does not have an effect here.**

Line 120: Given your study you need to explain what was covered in that paper, at least in brief.

**Answer:**

**We have included a brief outline of what was covered in that paper. The new sentence reads as follows "On two opposing hillslopes of a semi-arid, Mediterranean catchment in southern Spain, the study of the interaction between hydrology, terrain, and vegetation has been performed through soil moisture, vegetation, and water table dynamics measurement, to quantify the aspect influence on ecohydrological dynamics of an oak-woodland savanna or dehesa".**

Figure 1: neither summit looks like a summit...where is your stable pedon?

**Answer:**

**The geomorphological setting is a plateau with an incised river valley. Both profiles SC4 and SC10 are on the flat, stable part. This can be better appreciated in figure 6, and is very clear on the ground, as can be seen in the following pictures:**

[Figure]

*View of the SC4 location.*

[Figure]

*View of the SC10 location.*

Figure 1: need to note aspects on map

**Answer:**

**We have updated Figure 1 and noted aspects on map.**

Figure 1: why this specific...using a decimal for elevation?

**Answer:**

**We have updated Figure 1.**

Line 140: You need to present morphologic information on each pedon and physical and chemical characteristics.

**Answer:**

**We proposed to include this information as supplementary material because it is quite extensive. The morphologic information and the table with physical-chemical characteristics of the profiles are shown below:**

**Profile SC4**
**Description of the horizons**

| Depth (cm) | Horiz. | Description |
|---|---|---|
| 0-18 | A | Colour 7,5 YR 4/3; Sandy clay; structure granular, size very fine to fine; soft; few pores, very fine, irregular; few roots, very fine, distributed throughout the horizon; boundary abrupt and smooth. |
| 18-44 | B | Colour 10 YR 5/8; Sandy clay; structure granular, size very fine to fine; soft; few pores, very fine, irregular; few roots, very fine to coarse, distributed throughout the horizon; boundary gradual to wavy; 2% igneous rocks fragments partly weathered, discoidal and subangular, size gravel; mottles of krotovina 2-3 cm diameter; preferential routes of infiltration. |
| 44-110 | C | Colour 10 YR 6/6; unweathered parent material with oxidation of manganese, some yellow-grey mottles, abundant dark Mn mottles. |

**Profile SC5**
**Description of the horizon**

| Depth(cm) | Horiz. | Description |
|---|---|---|
| 0-18 | A | Colour 7,5 YR 3/4; Sandy loam; structure subangular, size very fine to coarse; soft; few pores, very fine, irregular; common roots, very fine to very coarse, distributed throughout the horizon; boundary abrupt and wavy; 2% igneous rocks fragments, subprismoidal and angular, fine-medium gravel. |
| 18-60 | B | Colour 5 YR 4/6; Sandy loam; structure granular, size medium to coarse; soil loose; no pores; common roots, very fine to very coarse, distributed throughout the horizon; boundary abrupt and wavy; horizon very weathered, no igneous rocks fragments. |
| 60-100 | C | Colour 10 YR 5/6; unweathered parent material; mottles of mangenese; very few roots, very fine to very coarse, between peds. |

**Profile SC6**
**Description of the horizon**

| Depth (cm) | Horiz. | Description |
|---|---|---|
| 0-55 | A | Colour 7,5 YR 2,5/2; Sandy loam; structure granular, size médium to coarse; soft; moderatly few pores, very fine to medium, irregular; common roots, very fine to very coarse, distributed throughout the horizon; boundary clear and wavy; 2% igneous rocks fragments, discoidal and angular, gravel to cobbles. |
| 55-95 | B | Color 7,5 YR 4/3; loamy sand; structure granular, size medium to very coarse; slightly hard; moderatly few pores, very fine to medium, irregular; moderatly few roots, very fine to very coarse, distributed throughout the horizon; 20-25% igneous rocks fragments, subidscoidal and angular, gravel to boulders. |

**Profile SC7**
**Description of the horizon**

| Depth (cm) | Horiz. | Description |
|---|---|---|
| 0-45 | A | Colour 7,5 YR 2,5/2; loamy sand; structure subangular, size fine to medium; soft; moderatly few pores, very fine to medium, irregular; common roots, very fine to very coarse, distributed throughout the horizon, boundary diffuse and smooth; 5-10% igneous rocks fragments, discoidal and very angular, gravel. |
| 45-97 | B | Colour 7,5 YR 4/3; loamy sand; structure granular, size medium to very coarse; soft; moderatly few pores, very fine to medium, irregular; moderatly few roots, very fine to coarse, distributed throughout the horizon; boundary gradual and wavy; 15-20% igneous rocks fragments, discoidal and very angular, gravel to stones; two large stones, spherical and subangular, the first (30x25 cm) between 45-60 cm depth, the second (40x22 cm) between 60-83 cm depth. |
| 97-160 | C | Colour 2,5 Y 5/3; sand; soil loose; no pores; very few roots, medium, between peds; very weathered horizon, igneous rocks fragments laminated; mottles black and orange. |

**Profile SC8**
**Description of the horizon**

| Depth (cm) | Horiz. | Description |
|---|---|---|
| 0-14 | A | Colour 7,5 YR 3/1; loamy sand; structure subangular, size fine to medium; slightly hard; few pores, very fine to very coarse, irregular; common roots, very fine to very coarse, distributed throughout the horizon; boundary clear and wavy; 15% igneous rocks fragments, discoidal and very angular, coarse gravel. |
| 14-47 | B | Colour 10 YR 5/6; Sandy loam; structure subangular; size medium to coarse; slightly hard; few pores, very fine to very coarse, irregular; common roots, very fine to very coarse, distributed throughout the horizon; boundary abrupt and irregular; 2-5% igneous rocks fragments, discoidal and very angular, medium gravel. |
| 47-240 | C | Colour 7,5 YR 5/8; sand; moderately hard; few pores, very fine to very coarse, irregular; common roots, very fine to very coarse, distributed throughout the horizon; slightly weathered igneous rocks fragments, coarse to very coarse, manganese coating between the fragments. |

**Profile SC9**
**Description of the horizons**

| Depth (cm) | Horiz. | Description |
|---|---|---|
| 0-19 | A | Colour 7,5 YR 2,5/3; Sandy loam; structure subangular, size medium to coarse; slightly hard; few pores, very fine, irregular; common roots, very fine to coarse, distributed throughout the horizon, boundary abrupt and wavy; 2-5% igneous rocks fragments, discoidal and very angular, size gravel to cobbles. |
| 19-57 | B | Colour 7,5 YR 4/6; loamy sand; structure subangular; size very fine to fine; soft; few pores, very fine, irregular; common roots, very fine to coarse, distributed throughout the horizon; boundary abrupt and irregular; 2-5% igneous rocks fragments, discoidal and very angular, fine gravel, |
| 57-137 | C | Colour 7,5 YR 6/8; sand; structure massive; size medium; slightly hard; few pores, very fine, irregular; common roots, very fine to coarse, distributed throughout the horizon; very weathered igneous rocks fragments (parent material). |

**Profile SC10**
**Description of the horizons**

| Depth (cm) | Horiz. | Description |
|---|---|---|
| 0-12,5 | A | Colour 7,5 YR 3/4; loamy sand; structure subangular, size medium to coarse; soft; few pores, very fine, irregular; common roots, very fine to fine, distributed throughout the horizon; boundary very abrupt and wavy; 0-2% igneous rocks fragments, discoidal and very angular, gravel. |
| 12,5-51 | B | Colour 7,5 YR 7/8; Sandy clay; structure massive, size medium to coarse; moderately hard; few pores, very fine, irregular; few roots, very fine to coarse, distributed throughout the horizon. |
| 51-73 | C1 | Colour 7,5 YR 5/6; sand; structure massive; unit soil loose; few pores, very fine, irregular; few roots, very fine to coarse, distributed throughout the horizon; boundary gradual and smooth; 0-2% igneous rocks fragments, discoidal and very angular, gravel, clay accumulation surfaces above fragments, accumulation of roots in water channels. |
| 73-100 | C2 | Colour 7,5 YR 6/8; sand; structure massive; unit soil loose; few pores, very fine, irregular; few roots, very fine to coarse, distributed throughout the horizon; 0-2% igneous rocks fragments, discoidal and very angular, parental material; clay accumulation surfaces above fragments, preference routes roots accumulation presence of water canals. |

| Profile | Profile Photo | Horizon | Depth cm | $pH_{1:2.5}$ | CaCO$_3$ % | OC% | Exchangeable cations (cmol$_{(+)}$kg$^{-1}$) | | | | CEC cmol$_+$ kg$^{-1}$ | Sand | Silt | Clay |
|---|---|---|---|---|---|---|---|---|---|---|---|---|---|---|
| | | | | | | | Ca | Mg | Na | K | | | % | |
| SC4 |  | A | 0-18 | 5.7 | 0 | 1.3 | 6.9 | 1.1 | 0.38 | 0.24 | 8.7 | 73.2 | 20.9 | 5.9 |
| | | B | 18-44 | 5.9 | 0 | 0.28 | 6.2 | 1.3 | 0.36 | 0.15 | 7.9 | 73.7 | 19.5 | 6.8 |
| | | C | 44-110 | 6.1 | 0 | 0.1 | 5.8 | 1.9 | 0.37 | 0.12 | 8.3 | 77.1 | 15.7 | 7.2 |
| SC5 |  | A | 0-18 | 6.0 | 0 | 0.66 | 8.7 | 1.6 | 0.4 | 0.2 | 10.9 | 75.7 | 18.1 | 6.2 |
| | | B | 18-60 | 6.2 | 0 | 0.19 | 11.0 | 2.4 | 0.4 | 0.15 | 14.0 | 85.7 | 9.6 | 4.7 |
| | | C | 60-100 | 5.9 | 0 | 0.12 | 10 | 3 | 0.5 | 0.15 | 13.6 | 86.4 | 9.9 | 3.7 |
| SC6 |  | A | 0-55 | 6.6 | 0.1 | 0.7 | 10.5 | 2.6 | 0.5 | 0.2 | 13.7 | 69.5 | 22.4 | 8.1 |
| | | B | 55-95 | 7.4 | 0.05 | 0.2 | 9.2 | 1.7 | 0.7 | 0.1 | 11.7 | 74.9 | 17.8 | 7.3 |
| SC7 |  | A | 0-45 | 6.3 | 0 | 0.9 | 9.2 | 2.3 | 0.9 | 0.2 | 12.7 | 73.3 | 17.9 | 8.8 |
| | | B | 45-97 | 6.7 | 0 | 0.3 | 8 | 2.6 | 0.9 | 0.1 | 11.7 | 78.9 | 13.3 | 7.8 |
| | | C | 97-160 | 7.0 | 0 | 0.1 | 12.4 | 5.1 | 0.6 | 0.1 | 18.3 | 74.8 | 12.6 | 12.6 |
| SC8 |  | A | 0-14 | 6.21 | 0 | 1.6 | 12.3 | 0.98 | 0.7 | 0.35 | 14.4 | 75.7 | 18 | 6.3 |
| | | B | 14-47 | 6.4 | 0 | 0.3 | 20.4 | 3.4 | 0.7 | 0.2 | 24.7 | 56.5 | 32.6 | 10.9 |
| | | C | 47-240 | 6.1 | 0 | 0.2 | 20.4 | 5.6 | 0.8 | 0.2 | 27 | 52.4 | 35.7 | 11.9 |
| SC9 |  | A | 0-19 | 5.8 | 0 | 1.5 | 10.3 | 2.4 | 0.7 | 0.3 | 13.7 | 71.7 | 22.4 | 5.9 |
| | | B | 19-57 | 6.3 | 0 | 0.3 | 9.1 | 2.1 | 0.7 | 0.2 | 12.2 | 76.1 | 18.4 | 5.5 |
| | | C | 57-137 | 6.7 | 0 | 0.1 | 8.2 | 1.9 | 0.7 | 0.2 | 11 | 81.1 | 13.6 | 5.3 |
| SC10 |  | A | 0-12.5 | 5.8 | 0 | 2.75 | 14.4 | 1.6 | 0.6 | 0.5 | 6.3 | 68.5 | 24.9 | 6.6 |
| | | B | 12.5-51 | 6.2 | 0 | 0.24 | 9.5 | 1.3 | 0.7 | 0.3 | 11.8 | 84.2 | 12.1 | 3.7 |
| | | C1 | 51-73 | 6.6 | 0 | 0.18 | 9.2 | 1.8 | 0.7 | 0.2 | 11.9 | 86.3 | 11.2 | 2.5 |
| | | C2 | 73-100 | 6.7 | 0 | 0.1 | 8.13 | 2.7 | 0.6 | 0.2 | 11.6 | 89.5 | 8 | 2.5 |

Line 145:Typo

**Answer:**
**We have corrected typo in line 145.**

Line 149:what kind of sensors?

**Answer:**
**We have included the kind of sensor used in line 149.**

Line 150:So you modeled the data for year 2 from year 1?
**Answer:**
**Soil moisture sensors started collecting data in November 2016 and since the initial soil moisture profile was unknown, precipitation and evapotranspiration data from 2017 was used for the rest of 2016 (January- October), due to data availability.**

Table 2: Too much unpublished data here...Are these theses?
**Answer:**

**No, these are not theses but were explicitly measured data, they are just not published since no journals publish simple profile or vegetation descriptions.**

What is the long-term dust deposition relationship in this area...CDPs are poor predictors in areas with lots of dust deposition from loess or eolian sources.

**Answer:**
**Firstly, absolute rates of dust deposition show that it is not very significant for soil formation in this area of Spain. Measurements by Vincent et al. (2016) indicate that current dust deposition rates in coastal areas vary between 1 and 7.4 g m2 year-1. Assuming a bulk density of 1.5 g cm-3, this translates into 0.066 to 0.49 cm per 1000 years only. As our study area is relatively far inland, rates are probably closer to the minimum.**
**Secondly, mineralogical studies indicate that only a fraction of the dust deposition is from Saharan dust with a different composition. Erel and Torrent (2010) suggested that Fe-oxides and Al-silicates in the fine (<5 µm) fraction of soils in southern Spain came in from two sources, i.e., (33-86%) Saharan dust and material released by the weathering of the basement, granitoid-type rocks. Liu et al, 2016 claimed that geochemical and clay mineral analyses indicated that aeolian dust significantly contributes to the A and B horizon material of the Spanish soil. However, the fraction of fines in the profiles in this study area is low. In addition, locations along the catena present a similar type of rock (granitoid material) so therefore this effect should not be very dramatic, as it will merely lead to a "dampening" of the differences created by weathering.**

An English editor should be used to proof the MS.

**Answer:**
**We will pay for a professional English language edition.**

Line 184: Was a standard method followed?

**Answer:**
**We have further elaborated on this section in the revised manuscript (lines 180-185).**

**The X-Ray Fluorescence (XRF) analyses were performed by the Scientific and Technical Services of the Universidad de Oviedo (Spain) following a standard method.**

**Results and discussion**

Line 203:Why is 1 yr of data enough for validation?

**Answer:**
**One year of data is used for validation because only one year of measurements was available at the time this work was performed.**

Figure 3: are these all data points here?

**Answer:**
**Yes, they show model results.**

Line 218: on same aspect, which was??

**Answer:**
**We have included this information in the revised manuscript. The new sentence reads as follows "The highest chemical weathering was measured in the profiles SC7 and SC8, on the NFS".**

Line 222: There must be others who propose different metrics of evaluation with CDFs...why just report this one paper's?

**Answer:**

We thought it was enough to cite only one paper to indicate a source for the criterion we used, but we have added this sentence: "The statistical metrics that are used for the quantitative evaluation of the results were also adopted by other authors such as Kontos et al. (2021) or Boylan et al. (2006)."

Line 228:This sentence does not follow the previous.

**Answer:**
**The sentences were reorganized and read as follows:**

**"A negative value of FB indicates model overestimation whereas RMSE and NMSE do not account for over or underestimation but their ideal value is zero (Brancher et al., 2020). The model, therefore, represents the measured trend in CDF values correctly, although there exists a positive bias. Note that the metric results shown are dimensionless because CDF is a non-dimensional quantity."**

Line 234:opposite aspect. Is it highly weathered or have dust deposition of highly weathered material??

**Answer:**

**We thank the reviewer for highlighting this sentence. In fact, it is an error on our part, it should be very little weathered, corresponding to the low CDF. The additional XRD analysis we did also confirms that rock composition around SC5 is different, which confirms the different weatherability. Also, the knickpoint in the surface topography confirms this. As mentioned before, we don't think current data shows dust deposition to be a significant factor in our study area.**

Line 238:why??

**Answer:**
**Based on Oesser et al., (2017) the high degree of weathering (CDF ≈ 0.4-0.5) despite low precipitation can be attributed to the low abundance of quartz on the one hand and the high abundance of weatherable plagioclase and mafic minerals on the other hand.**

Line 239: This is not clear...do you mean the soil composition is different from the rock? If so, is it weathering or dust inputs?

**Answer:**
**We meant that the mineralogical composition of the rock at location SC5 was different, which is corroborated by the XRD analysis we performed as suggested by the referee.**

Line 240:why does this exist? Differential weathering from two different lithologies?
**Answer:**
**That is our hypothesis.**

Line 243:Given no mineralogy data is presented I am not sure how you can make this claim...chemistry alone cannot answer your question.

**Answer:**
**Based on the results of the additional XRD analysis performed, both SC8 and SC5 rock samples stand out for the lower amount or lack of Mica, lower Quartz content, and higher Chlorite-Vermiculite content which seems to support the results of the chemical composition analysis of Table 3.**

Line 247:8 and 5 are on opposite aspects...I think you need to consider your basic geomorph and solar radiation effects. Which direction is the prevailing wind too...
**Answer:**

**Our previous study of soil moisture showed that solar radiation effects were counterbalanced by transpiration (higher on NFS), so soil moisture conditions were very similar on both slopes. Prevailing wind could be important for dust deposition, which we mentioned before, to be not significant here, or**

for orienting rainfall, but the study area is one of the most windless areas in Spain. We made rainfall measurements on opposing slopes and the records of the 5 rain gauges installed in the cross-section of the valley do not present any appreciable orography-induced differences.

Figure 6:What is the slope at each location...can you generate a 9-component slope model?

Answer:
Slope is given in Table 1. If we understand the comment correctly, it was generated by a 9-component slope model, as we calculated Slope in ArcGIS 10.5: "For each cell, the Slope tool calculates the maximum rate of change in value from that cell to its neighbors. The maximum change in elevation over the distance between the cell and its eight neighbors identifies the steepest downhill descent from the cell."In addition, we have completed Table 1 by performing a curvature analysis. Profile curvature affects the acceleration and deceleration of flow across the surface and planform curvature relates to the convergence and divergence of flow across a surface.

| | SC4 | SC5 | SC6 | SC7 | SC8 | SC9 | SC10 |
|---|---|---|---|---|---|---|---|
| Landform types | Hilltop | Mid Slope | Valley bottom | Valley bottom | Mid Slope | Upper Slope | Hilltop |
| Slope (°) | 4 | 20 | 14 | 4 | 27 | 19 | 2 |
| Upslope bearing (°) | 35 | 27 | 40 | 55 | 61 | 66 | 53 |
| Profile curvature ($m^{-1}$) | -0.47 | -0.22 | 2.5 | 1.0 | -3.0 | 0.23 | -0.75 |
| Plan curvature ($m^{-1}$) | 0.78 | 0.73 | -0.39 | -0.9 | 4.24 | 0.67 | -0.65 |

Figure 6:Why no errors bars if this is modeled?

Answer:
The SoilGen model is quite complex to run, and it takes several weeks to complete 1 simulation. The results shown here are therefore the result of 1 simulation. Although theoretically, we could do a sensitivity analysis, to calculate the error caused by the uncertainty of the different input factors, the model is simply too expensive to run.

Line 286: not necessarily...could be geomorphology...landforms and not slope...could be dust effects...could be a restrctive layer is shuttling water laterally...could be lots of things.
Answer:
We completely agree it could be different things. We have added some other possibilities like geomorphology or landforms. As explained before however, we don't believe dust effects are important here, nor restrictive layers, as the texture is too coarse for that.

Lines 286-288: do not follow this reasoning?

Answer:
The slope gradient is kept constant for the entire soil profile simulation time of 20000 years as we assume steady-state topography, following for instance Rempe and Dietrich (2014, PNAS). In 20000 years the slope gradient of the sample points could change. This is a common problem in historical modelling of soil erosion in agricultural landscapes as the slope diminishes with time due to erosion and the flattening out of the landscapes. In that type of context, the steady-state assumption is not valid, as erosion rates are quite high, but in any case, erosion is modelled using current topography because of a lack of a better alternative.

Line 294: There is lots of pedologic research going back 50+ years that has shown this too...I suggest you note that as well.

**Answer:**
**Agreed and changed.**

Line 295: why just natural?

**Answer:**
**Changed the sentence.**

Line 302: Again...you are implying first in terms of CDF modeling? This is old news for pedology.

**Answer:**

**Agreed if we talk about soil properties etc, but not in terms of CDF, or at least we could not find papers with data on this. We would be very happy to include them. We reviewed the papers from the review by Schaller and Ehlers (2021). Many authors analyze climate gradients over large distances, but not over valleys or catenas.**

Line 302: you have two different aspects though?

**Answer:**
**Yes, we have two different aspects but the records of the 5 rain gauges installed in the cross-section of the valley do not present any appreciable orography-induced differences as can be observed in the comparison of the daily rainfall in the Figure. The approximation to line 1:1 allows the estimation of the rainfall from anyone of the rain gauges in the rainy days, independent of the misfunctions of the other ones due to accidental blockage of the reception are by fallen leaves.**

[Figure]

*Comparison of daily rainfall, $p_{scx}$, recorded in the rain gauges at different locations, SCx.*

Lines 302-303: infiltration is about precipitation excess or surface condition (aggregate stability, cover, etc.)???

**Answer:**
**Our field observations lead us to believe surface condition does not have a big influence here.**

Line 310:was their topography similar to yours though?
Answer:
**No, they did not even study a catena, their study was more a landscape-scale study of different profiles in areas of different vegetation. We merely mention this study to indicate vegetation is important, as also indicated by the reviewer in the following comment.**

Line 313:more soil organic matter then...more acids...more mineralization??

Answer:
**We agree with this comment, vegetation could be an additional factor. We have added this to the text.**

Line 322:I think you have too many unknowns to claim this.

Answer:
**We have changed "can" to "could". We don't claim it is the only explanation, but we strongly believe this is the most likely explanation, taking into account also previous research from García-Gamero et al. (2021).**

Lines 323-324: but how do you know this?

Answer:
**To describe subsurface connectivity, researchers have come up with a number of different 'assessment tools' (Blume and van Meerveld, 2015). Most existing indicators are based on detailed measurements and combining shallow soil moisture patterns, surface and subsurface topography, although even with this data it is extremely complicated to come up with an adequate quantitative descriptor of connectivity as a tool for describing hydrological behaviour (Ali and Roy, 2010), let alone chemical weathering response. Therefore, in this study, while our field work based on soil moisture sensor data and aquifer response, clearly indicates the existence of significant subsurface connectivity on the NFS, and although we propose it to be an important reason for the deviation between modelled and measured CDF values, it is difficult to quantify its importance, let alone model it.**

Lines 326-327:yes it could, right across the bedrock surface.

Answer:
**We believe that it is unlikely a seasonal water table develops over a bedrock that is at 60 cm depth and that water could travel over a distance of 200 m this way. This would create local seepage areas at any change in slope or bedrock outcrop area on the SFS, that could be easily observed in the field. This was not the case, nor did we observe seasonal groundwater in soil profiles or augering during the winter season. On the NFS on the other hand, we had both.**

Line 327:what is the microlandform...the soil horizons?

Answer:

**We have included this information in supplementary materials. See above.**

Line 334:precipitation would not be equivalent in effect on the 2 aspects.
Answer:

**No, this sensitivity analysis was only performed for the SC10 location at the flat area because of the beforementioned model cost.**

Line 334-339: You need to include your morphology data to really know if this is working as you claim.
Answer:
**Included in supplementary materials. See above.**

I want to see the profile data and hear the answers to my prior questions before considering any of the remaining text.

**Conclusions**

I want to see the prior questions addressed before reviewing these conclusions.

**RC3: 'Comment on soil-2021-78', Anonymous Referee #3, 15 Nov 2021**

Review of the manuscript "Modelling the effect of catena position and hydrology on soil chemical weathering" by García-Gamero et al.

This research work presents application of a 1dimentional pedogenetic (soil profile evolution) model SoilGen on various points on a catena sale landform and comparing the model results to the measured soil properties in the catena. Through this the authors attempt to identify links between topographic position and hydrological attributes with the chemical weathering of soil profiles in said landform. In addition, a sensitivity analysis of the model has been done using a range of annual average precipitation.

From this work the authors have found that the chemical weathering (represented by chemical depletion fraction (CDF)) seems to have no or very weak correlation with catena position or the slope gradient but seems to have some correlation with hydrological factors such as soil moisture and infiltration. The sensitivity analysis of the model has shown some interesting relationship between the annual average precipitation and chemical weathering. Increasing annual average precipitation seems to increase chemical weathering up to a threshold value after which chemical weathering rate seems to decrease.

The manuscript is well written and easy to follow. The results are well presented and the results interpretation and the physical underpinnings of the results are well described. However I have some concerns with the manuscript (described below) that will probably amount to minor revisions. I believe that the manuscript is acceptable for publication after these concerns have been addressed.

1. I believe the readers would benefit form an additional section to the manuscript describing the SoilGen model and its physical underpinnings. The mathematical formulations of such a model maybe too complex to be describe in detail in manuscript like this. However a brief description on how the model works (maybe with a flow diagram) would complete the manuscript.

**Answer:**
**We appreciate the suggestion of the reviewer and have included a brief description of the model and flowchart:**
**SoilGen (Finke, 2012; Finke and Hutson, 2008) simulates the change of soil properties as a function of properties of the parent material and time-dependent drivers at the soil boundary (climate, vegetation, bioturbation, relief and deposition or erosion). The model operates at a typical spatial scale of 1 m2, covers millenniums but takes dynamic time steps that vary per process (Figure 1), depending on process speed. The flow of water, heat, gas and solutes is represented by numerical solutions to partial differential equations (Richards' equation, heat flow equation, gas diffusion equation, solute advection/dispersion equation), where soil profiles have 5-cm compartments. For water flow, the relationship between pressure head, water content and hydraulic conductivity are dynamically parameterized using a pedotransfer function based on the texture, organic matter content and bulk density. These properties are dynamically simulated per compartment: (i) The fate of organic carbon is simulated according to the concepts of the RothC26.3 model (Coleman and Jenkinson, 19966); (ii) The texture changes due to physical weathering of minerals; (iii) Clay migration and bioturbation affect the vertical distribution of all soil components; (iv) The bulk density varies because of mass gains/losses over the compartment.**

**Chemical weathering of minerals as well as organic matter decomposition release ions in the soil solution. These ions are distributed over precipitated, solution and exchange phases using a Gapon exchange mechanism and chemical equilibriums. The model can simulate several agricultural practices**

**and can also accommodate the removal (or addition) of top layers by erosion (or sedimentation, such as dust addition).**

[Figure]

*Process flowchart of SoilGen. From Finke, 2012.*

2. In this work, authors have used slope gradient as the only topographical variable in their analysis. Slope does indeed influence the hydrological state of different areas of the catchment. It is well known that both slope and upstream contributing area (cumulative area of the catchment which drain through a particular point) determine the hydrological state of a node in a catchment. In fact all the landform/soilscape evolution models (SIBERIA, mARM3D, SSSPAM) in 3 dimension use area and slope as primary variables for erosion calculations. As I understand the sample points does not particularly lie on a transect or on the same drainage line. So the upstream contributing area could be very different sample point to point. I believe that an analysis on the relationship of contributing area (or the combination of slope and area) and the CDF could be beneficial to this manuscript.

**Answer:**
**We analyzed the relationship between contributing area and the calculated CDF but was not significant with an $R^2$= 0.14 and a p-value >0.05.**

[Figure]

3. As I understand the slope gradient is kept constant for the entire soil profile simulation time of 20000 years? In 20000 years the geomorphology of the catchment would most definitely change and the slope gradient and (for a lesser extent) contributing area of the sample points would also change. The authors have noted this issue and have attributed the difference in the observed and simulated CDF values for this changes. Were any geomorphological evolution (landform evolution) simulations of the catena done using available landform evolution model such as MILESD, LORICA, or SIBERIA to characterise the change of geomorphological attributes (slope gradient and/or contributing area) of the sample points over the simulation time of 20000years? If the authors have done such simulation work they could use a time series of slope gradients extracted from the landform evolution simulation results at each sample point and input that into their soil profile simulation model which may give better results.
**Answer:**
**Topography evolution in long-term simulation is a common problem in this type of works because the topographic initial conditions are unknown. It is common to assume steady-state conditions, following Rempe and Dietrich (2014), although we are aware of the constraint this involves.**

4. Page 9: Figure 2: This figure shows the time series input variables used for the SoilGen model over 20000years of simulations. According to the figure 2 form 12000years to 20000years the input variables seems to be constant. Is this an effect of data not being available beyond 12000years in to the history and average values were used from 12000 to 20000 years or the variables are found to be constant using any other means? This was not clear from the text describing the figure2.

**Answer:**
**The climate reconstruction is based on the work of Davis et al.,2003, which is available for the last 12,000 years, and updated by Mauri et al. (2015, Quaternary Science Reviews). To the best of our knowledge, there are no longer records available at this resolution needed for the model and that includes the climate variables required in this model (for example Jiménez de Cisneros and Caballero, 2013. Natural Science, that only has temperature based on speleothems). Locally, some longer climate reconstructions**

**are available but not representative of our area, or without quantitative climate reconstruction data (for example Jiménez-Moreno et al., 2019. Global and Planetary Change).**

**In any case, a previous study by Keyvanshokouhi et al. (2016, Science of the Total Environment) analyzed how uncertainty in the boundary conditions affected the results of the Soilgen model and concluded that climate data uncertainty at the beginning of the series did not affect results significantly. Thus, our climate data uncertainty between 20.000-12.000 years BP is (1) the best available data we could find and (2) we are confident a possible error here would not have a significant impact on the results.**

5. Page 18: Figure 8: In this figure the authors have provided the results of the sensitivity analysis of the model with respect to different annual average precipitations. The CDF of the upper soil layers (<40cm) increase with precipitation until 800mm/year and then decrease. The general pattern of the CDF-soil depth variation is consistent for all the CDF-depth curves where the CDF is high in shallow soil compared to deeper soil, except for the 900mm curve where the CDF of the upper soil layers are lower than the deep soil layers. The 1000mm curve seems to show the same trend as well. It would be beneficial to the readers if the authors could elaborate on this "inversion" of depth vs chemical withering curve at 900mm and the physical underpinnings they suspect leading to this anomaly.

**Answer:**
**The reviewer is right in observing that the depth patterns of the CDF change when the precipitation exceeds 900 mm. We consider this a function of the absence or presence of a precipitation surplus. Our explanation is as follows: In the situation with low precipitation (P=200-600 mm/y), the rainfall is lower than the potential evapotranspiration, and as a consequence, only the upper 45 cm of the profile are wetted while the lower part remains dry. This means that the contact time of meteoric water with the topsoil is long while it is near 0 in the subsoil. A long contact time allows for more weathering and this is reflected by higher CDF in the topsoil than in the subsoil. This contrast increases with P increasing from 200 to 600 mm. At P of 800 mm the lower part of the profile also is moistened and the contrast decreases, while the contact time is still large, resulting in a high CDF. At even higher P, the subsoil is moistened and this increases the hydraulic conductivity and thus decreases the contact time. As a result, the CDF decrease, and also the topsoil-subsoil contrast in CDF disappears.**